# ResAD: A Simple Framework for Class Generalizable Anomaly Detection

**Xincheng Yao[1], Zixin Chen[1], Chao Gao[3], Guangtao Zhai[1], Chongyang Zhang[1,2*]**

[1]School of Electronic Information and Electrical Engineering, Shanghai Jiao Tong University
[2]MoE Key Lab of Artificial Intelligence, AI Institute, Shanghai Jiao Tong University
[3]China Pacific Insurance (Group) Co., Ltd.
`{i-Dover, CZX15724137864, zhaiguangtao, sunny_zhang}@sjtu.edu.cn`[1]
`gaochao-027@cpic.com.cn`[3]

## Abstract

This paper explores the problem of class-generalizable anomaly detection, where the objective is to train one unified AD model that can generalize to detect anomalies in diverse classes from different domains without any retraining or fine-tuning on the target data. Because normal feature representations vary significantly across classes, this will cause the widely studied one-for-one AD models to be poorly class-generalizable (*i.e.*, performance drops dramatically when used for new classes). In this work, we propose a simple but effective framework (called ResAD) that can be directly applied to detect anomalies in new classes. Our main insight is to learn the residual feature distribution rather than the initial feature distribution. In this way, we can significantly reduce feature variations. Even in new classes, the distribution of normal residual features would not remarkably shift from the learned distribution. Therefore, the learned model can be directly adapted to new classes. ResAD consists of three components: (1) a Feature Converter that converts initial features into residual features; (2) a simple and shallow Feature Constraintor that constrains normal residual features into a spatial hypersphere for further reducing feature variations and maintaining consistency in feature scales among different classes; (3) a Feature Distribution Estimator that estimates the normal residual feature distribution, anomalies can be recognized as out-of-distribution. Despite the simplicity, ResAD can achieve remarkable anomaly detection results when directly used in new classes. The code is available at `https://github.com/xcyao00/ResAD`.

## 1 Introduction

Anomaly detection (AD) has achieved rapid advances in many application domains, such as industrial inspection, video surveillance, and medical lesion detection [9, 26]. However, applying AD algorithms in real-world scenarios still confronts many challenges. A critical challenge is that there are usually diverse classes[2] and new classes are continually emerging. Most previous one-for-one and also one-for-many (*i.e.*, learning one AD model for multiple classes) AD methods [13, 17, 43, 12, 30, 47, 46, 44] are still insufficient to satisfy the requirements of real-world applications. Because such methods still require retraining or fine-tuning when encountering new classes, but application users generally don't have such ability. Another more fatal point is that some scenarios may not allow retraining on target classes due to data privacy issues [41]. Therefore, the class-generalizable ability is a critical issue in the AD community, but it still hasn't been well studied in most AD literatures.

---

[*]Corresponding Author.

[2]Class means the category of the object in the image. For industrial scenarios, it refers to the product category, *e.g.*, bottle, carpet, etc. For medical analysis, it refers to the body organ category, *e.g.*, head CT, retina, etc.

38th Conference on Neural Information Processing Systems (NeurIPS 2024).

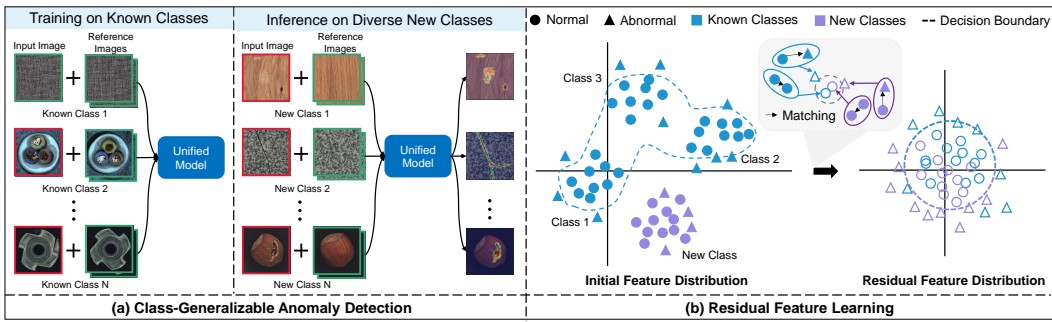

Figure 1: **(a)**: Intuitive illustration of class-generalizable anomaly detection **(b)**: Conceptual illustration of residual features. The residual feature space has fewer variations compared to the initial feature space. The decision boundary of the residual feature distribution can more effectively distinguish anomalies in new classes, rather than treating features of new classes as anomalies.

In this paper, we aim to tackle an academy-valuable and application-required task: few-shot class-generalizable anomaly detection, *i.e.*, one unified model is trained with samples from multiple known classes, and the goal is that the trained model can generalize to detect anomalies in new[3] classes without any retraining or fine-tuning on the target data, only few-shot new class normal samples are required. Nonetheless, solving such a task is quite challenging. The current one-for-one/many AD models have almost no ability to directly generalize to new classes. The main challenge is: *the normal patterns from different classes are significantly different.* This can lead to many normal misdetections of new classes, *i.e.*, normal patches from new classes may be mistaken as abnormal as they are quite different from the learned normal patterns. Thus, how to design a class-generalizable AD model under the feature variation circumstance? Our design philosophy is: "seeking invariation from variation". We think that residual features (*i.e.*, formed by subtracting normal reference features) can be regarded as class-invariant[4] representations compared to the significantly variant initial features. As shown in Fig.1(b), the main merit of normal residual features is: even in new classes, the distribution of normal residual features would not remarkably shift from the learned distribution. Regardless of classes, larger residuals are expected for abnormal features than normal features (please see Sec.3.1).

To this end, we propose a simple but effective class-generalizable AD framework, called ResAD (*i.e.*, **Res**idual Feature Learning based Class-Generalizable **A**nomaly **D**etection). ResAD is based on one key insight: residual feature learning, and consists of two key designs: feature hypersphere constraining and feature distribution estimating. First, we propose to use residual features for reducing class feature variations. We employ a pre-trained feature extractor to generate normal reference features from few-shot normal reference samples. Each input feature will match the nearest normal reference feature and subtract it to form the residual feature. In this way, the most variable class-related components are very likely to be mutually eliminated, resulting in residual features distributed in a relatively fixed origin-centered region (please see Sec.3.1). Second, to further reduce the variations in the residual feature space, we take the idea from the one-class-classification (OCC) learning [37, 22] to constrain the feature space. Specifically, we employ a simple and shallow network and propose an abnormal invariant OCC loss to transform normal residual features into a constrained spatial hypersphere. Third, with the hypersphere-constrained feature space, we can easily utilize a feature distribution estimator [14] to learn and estimate the normal residual feature distribution, anomalies can be recognized as out-of-distribution. For new classes, as the residual features have fewer variations or are covered by the learned distribution, the whole framework is more class-generalizable. Our contributions are as follows:

1. To accomplish class-generalizable anomaly detection, we propose a simple but effective framework: ResAD, which can be applied to detect and localize anomalies in new classes.

2. We are innovatively based on residual feature learning to address the issue of previous one-for-one/many AD methods not being able to generalize to new classes.

---

[3]We call the classes in training as known classes, others are called as new (or novel, unknown) classes.

[4]Strictly speaking, the residual features are not fully invariant, while the variation is smaller.

3. Comprehensive experiments on six real-world AD datasets are performed to evaluate the AD model's class-generalizable ability. With only 4-shot normal samples as reference, ResAD can achieve remarkable AD results, significantly outperforming the state-of-the-art competing methods.

## 2 Related Work

**One-for-One/Many AD Methods.** Most AD methods follow the one-for-one/many paradigm. (1) *Reconstruction-based methods* are the most popular AD methods. These methods hold the insight that models trained by normal samples would fail in abnormal image regions. Many previous works utilize auto-encoders [8, 27, 42], masked auto-encoders [46], variational auto-encoders [21] and generative adversarial networks [36, 1] to encode and reconstruct normal data. UniAD [47] is a transformer-based reconstruction model and mainly based on neighbor masked attention to address the "identical shortcut" issue to achieve one-for-many AD. (2) *Distillation-based methods* [6] can also be considered as belonging to the reconstruction type. These methods train student networks to reconstruct the representation of teacher networks on normal samples, and the assumption is that the student would fail in abnormal features. Recent works mainly focus on feature pyramid [35, 39], reverse distillation [13, 38], and asymmetric distillation [33]. (3) *Embedding-based methods* mainly rely on good feature representation and assume that abnormal features are usually far from the normal clusters. Most superior methods [11, 4, 12, 29, 30] utilize ImageNet pre-trained networks for feature extraction. However, industrial images generally have an obvious distribution shift from ImageNet. To better account for the distribution shift, subsequent adaptations should be done. The normalizing flow-based methods [31, 17, 32, 48, 44] are proposed to transform the pre-trained feature distribution into latent Gaussian distribution, and thus can better learn the normal data distribution. HGAD [44] proposes a novel hierarchical Gaussian mixture normalizing flow modeling method to address the "homogeneous mapping" issue for accomplishing one-for-many AD.

**Few-Shot AD Methods.** The few-shot AD methods have more similarities with ours. Distance-based approaches such as SPADE [11], PaDiM [12], and PatchCore [30] can be adapted to address few-shot AD by only making use of few-shot normal samples to calculate distance-based anomaly scores without training networks. RegAD [18] proposes to train a feature registration network to align input images and follows PaDiM [12] to model Multivariate Gaussian distribution with few-shot normal samples. The idea in FastRecon [15] is to reconstruct an anomalous sample to its normal version by few-shot support samples. A novel regression with distribution regularization is proposed to obtain the optimal transformation from support to query features. Recently, the CLIP-based AD methods, including WinCLIP [19] and VAND [10] show better few-shot AD performance. They both employ a text prompt ensemble strategy to obtain the language-guided anomaly map.

We think class-generalizable AD and few-shot AD are still not the same, they still have some differences. Class-generalizable AD requires the model to be class-generalizable, and we only extract features of normal samples in the new class as reference. Few-shot AD mainly focuses on how to effectively utilize few-shot normal samples to construct AD models, some dedicated modules may be introduced to handle the few-shot normal samples. These methods usually still need to re-model in new classes based on few-shot normal samples, *e.g.*, RegAD needs to re-model Multivariate Gaussian distribution for new classes. The CLIP-based methods can be seen as class-generalizable, as these methods can obtain anomaly maps by aligning vision features with text features. However, they heavily rely on the visual-language comprehension abilities of CLIP and handcrafted text prompts about defects, making them difficult to generalize to anomalies in diverse classes. Compared to the few-shot AD methods, our method can learn a class-generalizable AD model, which can be directly applied to new classes only requiring extracting features of few-shot normal samples as reference.

More recently, InCTRL [50] proposes to use few-shot normal images as sample prompts and learn to capture in-context residuals between the query image and sample prompts. The idea of in-context residuals in InCTRL is very similar to ours. But our method has obvious differences with InCTRL in the definition and utilization of residuals (please see the detailed differences in Appendix A.1).

## 3 Method

**Problem Statement.** In the class-generalizable AD task, we focus on the performance of new classes. Formally, let $\mathcal{I}_{train} = \mathcal{I}^n \cup \mathcal{I}^a$ be a training dataset with normal images and some anomalies (*i.e.*,

anomalies that exist in training set should also be effectively utilized), where $\mathcal{I}^n = \{I_i^n\}_{i=1}^N$ and $\mathcal{I}^a = \{I_j^a\}_{j=1}^M$ indicate the collection of normal samples and abnormal samples. As for testing, the model is evaluated on a collection of other AD datasets ($\mathcal{T} = \{\mathcal{I}_1^{test}, \mathcal{I}_2^{test}, \ldots, \mathcal{I}_T^{test}\}$) except the training dataset. The classes in the test set are drawn from unknown classes $\mathcal{C}_u$ that are different from the known classes $\mathcal{C}_k$ in the training set. Then the goal is to learn one unified model $\mathcal{M} : \mathcal{I} \to \mathbb{R}$ that is trained with known classes $\mathcal{C}_k$ and can directly adapt to unknown classes $\mathcal{C}_u$ without any retraining or fine-tuning on the target data (only few-shot (*e.g.*, 4) normal samples as reference).

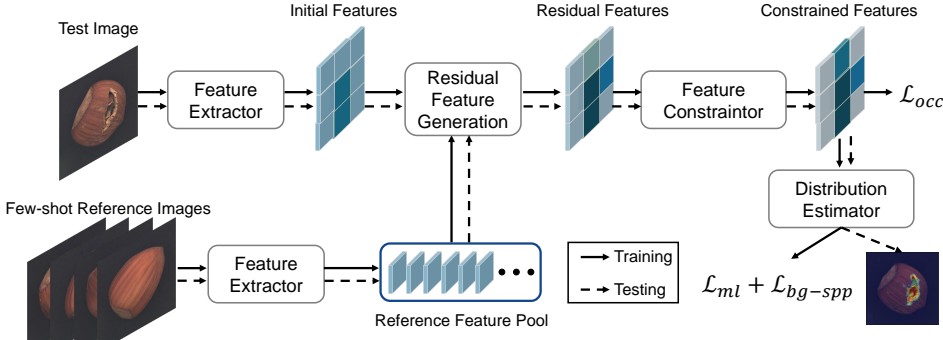

Figure 2: Framework overview. Note that the training samples belong to different classes. First, few-shot normal reference samples are fed into a pre-trained *Feature Extractor* to obtain normal reference features. Each initial feature will match the nearest normal reference feature and subtract it to form the residual feature. Then, a *Feature Constraintor* is utilized to transform the normal residual features into a constrained spatial hypersphere. Finally, we employ a normalizing flow model as the *Feature Distribution Estimator* to learn and estimate the residual feature distribution.

**Overview.** The proposed ResAD framework is illustrated in Fig.2. The ResAD framework consists of three parts: a *Feature Extractor*, a *Feature Constraintor*, and a *Feature Distribution Estimator*. These modules will be described below in sequence.

### 3.1 Residual Feature Generating

Residual feature learning is our core insight for solving class-generalizable anomaly detection. In this subsection, we describe how to generate residual features. For any input image $I_i \in \mathbb{R}^{H \times W \times 3}$, we follow the common practice of previous AD methods to employ a pre-trained feature extraction network $\phi$ to extract features from different levels. Formally, we define $L$ as the total number of levels for use. The feature map from level $l \in \{1, 2, \ldots, L\}$ is denoted as $\phi^l(I_i) \in \mathbb{R}^{H_l \times W_l \times C_l}$, where $H_l$, $W_l$ and $C_l$ are the height, width, and channel dimension of the feature map. For an entry $x_{h,w}^l = \phi^l(I_i)_{h,w} \in \mathbb{R}^{C_l}$ at level $l$ and location $(h, w)$, we will match it with the nearest normal reference feature from the corresponding reference feature pool, and then convert it into the residual feature. The details are described in the following:

**Reference Feature Pools.** The reference feature pools are utilized to store some normal features as reference. For new classes, we will provide few-shot normal samples (*i.e.*, randomly selected and then fixed, please see our discussion on sample selection in Appendix A.2) as reference. The pre-trained network $\phi$ will extract hierarchical features for these normal reference images, then the extracted features are sent into the feature pools as reference features. For $l$th level, the $l$th reference feature pool is composed of $\mathcal{P}_l = \{x_{h,w}^{l,i} | h \in \{1, \ldots, H_l\}, w \in \{1, \ldots, W_l\}, l \in \{1, \ldots, L\}, i \in \{1, \ldots, N_{fs}\}\}$, where $i$ denotes the $i$th normal sample, the $N_{fs}$ is the number of normal reference samples.

**Residual Features.** For each initial feature $x_{h,w}^l$, we can search the nearest nominal reference feature $x_n^* = \mathrm{argmin}_{x \in \mathcal{P}_l} ||x - x_{h,w}^l||_2$ from the $l$th reference feature pool $\mathcal{P}_l$. Then, we define the residual representation of $x_{h,w}^l$ to its closest normal reference feature as:

$$x_{h,w}^{l,r} = x_{h,w}^l - x_n^*. \tag{1}$$

**Why can residual features be less sensitive to new classes compared to initial features?** Because they are obtained by matching and then subtracting. From the principles of representation learning,

we know that features of each class generated by well-trained neural networks usually have some class-related attributes to the class for distinguishing from other classes [3]. The "class-related" means these attributes are typical to the class and distinctive from other classes, representing the most discriminative characteristics of the class. Thus, features from different classes are usually located in different feature domains [40]. However, as class-related attributes can also exist in normal reference features (they are usually in the same feature domain as the input query feature), the matching process can be seen as matching the most similar class-related attributes to each query feature. Therefore, by subtracting, the class-related components in the initial features are very likely to be mutually eliminated, leaving the highlighted discrepancy between normals and anomalies (*i.e.*, larger residuals are more likely to be anomalies than normal features). Thus, it can be imagined that the normal residual features generally will be distributed in an origin-centered region, even in new classes, the feature distribution region would not remarkably shift (please see the t-SNE visualization in Fig.3).

## 3.2 Feature Hypersphere Constraining

Even if the feature variations in the residual feature space will be significantly reduced relative to the initial feature space. Features of different classes may still have significant differences in scale, namely, the numerical value scales in the features of different classes may be remarkably different. This can lead to difficulty in obtaining a unified normal-abnormal decision boundary of different classes, *i.e.*, the scales of decision boundaries in different classes may be significantly different, a good decision boundary in one class may be poor in another class. In order to further reduce feature variations and also maintain the consistency in feature scales among different classes, we take the idea from one-class-classification (OCC) learning [34, 22] and propose a *Feature Constraintor* to constrain the initial normal residual features to a spatial hypersphere. The *Feature Constraintor* $C_{\theta_1}$ projects the initial residual feature $x_{h,w}^{l,r}$ to the constrained feature $x_{h,w}^{\prime,l,r}$ as $x_{h,w}^{\prime,l,r} = C_{\theta_1}(x_{h,w}^{l,r})$.

Because we only want to further reduce the variations in the initial residual distribution by constraining and don't want to change the distribution overly, we adopt a simple Conv+BN+ReLU layer as the network of our *Feature Constraintor*. A complex network may lead to overfitting known features, reducing the generalization ability for new classes (please see ablation studies in Tab.2(b)).

**Abnormal Invariant OCC Loss.** We propose an abnormal invariant OCC loss to optimize our *Feature Constraintor*. The loss is defined as:

$$\mathcal{L}_{occ} = \frac{1}{L} \sum_{l=1}^{L} \left( \frac{1}{H_l W_l} \sum_{h=1}^{H_l} \sum_{w=1}^{W_l} (1 - y_{h,w}^l) || \sqrt{||x_{h,w}^{\prime,l,r}||_2 + 1} - 1 ||_1 + y_{h,w}^l ||x_{h,w}^{\prime,l,r} - x_{h,w}^{l,r}||_2 \right). \quad (2)$$

where $y_{h,w}^l = 1$ denotes the $(h, w)$ position on the feature map is anomalous and $y_{h,w}^l = 0$ denotes a normal position (we can downsample the ground-truth mask to a low-resolution mask, which can indicate normal and abnormal positions). The first part in the loss function is a pseudo-Huber loss [22], which is used for constraining the normal residual features to a hypersphere. However, if we only constrain features to the hypersphere, the network may more easily overfit and simply map all features to the hypersphere. If we give the network another objective for anomalous features, this will urge the network to distinguish between normal and abnormal, rather than forming a shortcut solution. Thus, we further introduce an abnormal invariant term by simply predicting the initial features $||x_{h,w}^{\prime,l,r} - x_{h,w}^{l,r}||_2$. "Invariant" means the abnormal residual features remain relatively unchanged relative to themselves and will not be mapped to the hypersphere. In this way, our proposed abnormal invariant OCC loss can not only make the distribution of normal residual features more compact but also keep abnormal residual features as invariant as possible. In addition, by constraining normal features into a hypersphere, the normal feature scales of different classes can also be more consistent. Therefore, after the *Feature Constraintor*, the normal and abnormal residual features are more distinguishable (see Fig.3), namely, we can obtain a better unified decision boundary.

## 3.3 Feature Distribution Estimating

We employ the normalizing flow (NF) model [14] as our *Feature Distribution Estimator* to estimate the residual feature distribution. Note that our framework is not limited to normalizing flow, and other generative models can also be used as the distribution estimator. Formally, we denote $\varphi_{\theta_2}$ : $\mathcal{X} \in \mathbb{R}^{C_l} \to \mathcal{Z} \in \mathbb{R}^{C_l}$ as our NF model. The input residual feature $x_{h,w}^{\prime,l,r}$ will be transformed into a

latent feature $z_{h,w}^l = \varphi_{\theta_2}(x_{h,w}^{',l,r})$ by the NF model. The estimated residual distribution $p_{\theta_2}(x)$ can be calculated according to the change of variables formula as follows [14, 20]:

$$\log p_{\theta_2}(x) = \log p_Z(z) + \log|\det J|. \tag{3}$$

where the $J = \nabla_x z$ is the Jacobian matrix of the bijective transformation $\varphi_{\theta_2}$. The model parameters $\theta_2$ can be optimized by maximizing the log-likelihoods, and the latent variables $Z$ for normal features are usually assumed to obey $\mathcal{N}(0, I)$. The maximum likelihood loss function for learning normal residual feature distribution is derived as:

$$\mathcal{L}_{ml} = \frac{1}{L} \sum_{l=1}^{L} \left( \frac{1}{H_l W_l} \sum_{h=1}^{H_l} \sum_{w=1}^{W_l} \frac{C_l}{2}\log(2\pi) + \frac{1}{2}(z_{h,w}^l)^T z_{h,w}^l - \log|\det J_{h,w}^l| \right). \tag{4}$$

In the class-generalizable AD task, in addition to learning from normal samples, it's also valuable for us to effectively utilize abnormal samples that exist in known classes. Considering that we focus on detecting unknown anomalies in new classes, we cannot overfit the anomalies in known classes. Thus, following BGAD [45], we employ the explicit boundary guided semi-push-pull loss to learn a more discriminative and also generalizable feature distribution estimator. The loss is defined as:

$$\mathcal{L}_{bg-spp} = \sum_{i=1}^{N_n} |\min(\log p_i - b_n, 0)| + \sum_{j=1}^{N_a} |\max(\log p_j - b_n + \tau, 0)|. \tag{5}$$

where $b_n$ is an explicit normal boundary, $\tau$ is a margin, $N_n$ and $N_a$ denote the number of normal and abnormal features in a training batch. We set $b_n$ according to the way in BGAD, and $\tau$ is set to 0.1. Then, the whole loss function for training is as follows:

$$\mathcal{L} = \mathcal{L}_{occ} + \mathcal{L}_{ml} + \mathcal{L}_{bg-spp}. \tag{6}$$

In Appendix E, we further discuss the sensitivity of balancing among the three loss terms.

### 3.4 Inference and Anomaly Scoring

For new classes, our method only requires few-shot normal samples to extract features as reference, without any fine-tuning. We feed each test feature $x_i^l$ into the *Feature Constraintor* $C_{\theta_1}$ and the *Feature Distribution Estimator* $\varphi_{\theta_2}$ to get the latent feature $z_i^l$. The anomaly score is calculated as:

$$s(x_i^l) = 1 - \exp\left( -\frac{C_l}{2}\log(2\pi) - \frac{1}{2}(z_i^l)^T z_i^l + \log|\det J_i^l| \right). \tag{7}$$

Then, we upsample all $s(x_i^l)$ in the $l$th level to the input image resolution ($H \times W$) using bilinear interpolation and combine all levels (*i.e.*, sum) to obtain the final anomaly map. The maximum score of the anomaly map is taken as the anomaly detection score of the image.

## 4 Experiments

### 4.1 Experimental Setup

**Datasets and Metrics.** We conduct comprehensive experiments on four real-world industrial AD datasets, including MVTecAD [5], VisA [51], BTAD [25], and MVTec3D [7]. The detailed introduction to these datasets is provided in Appendix D. For MVTec3D, we only use RGB images in the dataset. As for our method's generalizability to other domains, we further evaluate our method on a medical image dataset, BraTS [24] (for brain tumor segmentation) and a video AD dataset, ShanghaiTech [23]. As our method is image-based, we extract video frames in ShanghaiTech as images for use.

Following previous works [5, 6], the anomaly detection performance is evaluated using the Area Under the Receiver Operating Characteristic Curve (AUROC).

To examine the model's class-generalizable ability, we evaluate the cross-dataset performance. We combine the training and test sets of the MVTecAD dataset to train AD methods, and they are subsequently evaluated on the test set of other five datasets without any retraining, *e.g.*, we train AD

models on MVTecAD and test on VisA. For MVTecAD, we train AD models on VisA. We report the performance with the number of few-shot normal samples set to $K = 2, 4$.

**Implementation Details.** All the training and test images are resized and cropped to $224 \times 224$ resolution. Following the common practice in AD literatures, we utilize the commonly used WideResNet50 [49] as the feature extractor, and the outputs from the $[1, 2, 3]$ layers of WideResNet50 are used as the pre-trained features. The parameters of the feature extractor are frozen during training. The layer numbers of the NF model are set as 8. We use the Adam [28] optimizer with weight decay $5e^{-4}$ to train the model. The total training epochs are set as 100, and the batch size is 32. The learning rate is $1e^{-5}$ initially and dropped by 0.1 after $[70, 90]$ epochs. During training, we randomly select reference samples for each input image to increase residual feature diversity. The network details are in Appendix B, we also evaluate the computation costs of our model and other competing models. We run all the experiments with a single NVIDIA RTX 4090 GPU and random seed 42.

Table 1: Anomaly detection and localization results with AUROC metric on six real-world AD datasets under various few-shot AD settings. ·/· means image-level and pixel-level AUROCs. RDAD and UniAD don't utilize the few-shot normal samples to fine-tune, so the results under 2-shot and 4-shot are the same. For each input image, InCTRL only outputs an image-level anomaly score. Thus, the pixel-level AUROCs of InCTRL are missing.

| Setting | Datasets | Baselines | | Few-shot AD Methods (Non-CLIP-based) | | | | ResAD | CLIP-based AD Methods | | ResAD† |
| | | RDAD CVPR2022 | UniAD NeurIPS2022 | SPADE | PaDiM | PatchCore CVPR2022 | RegAD ECCV2022 | (Ours) | WinCLIP CVPR2023 | InCTRL CVPR2024 | (Ours) |
|---|---|---|---|---|---|---|---|---|---|---|---|
| **2-shot** | | | | | | | | | | | |
| | Industrial AD Datasets | | | | | | | | | | |
| | MVTecAD | 65.9/71.9 | 67.4/81.1 | 74.6/64.0 | 79.5/93.8 | 74.7/85.2 | 80.4/93.3 | 85.6/94.1 | 93.1/93.8 | 94.0/- | **94.4/95.6** |
| | VisA | 56.4/79.9 | 52.1/81.8 | 71.7/65.4 | 68.7/91.5 | 65.0/80.4 | 70.6/93.3 | 79.9/**96.4** | 81.9/94.9 | **85.8**/- | 84.5/95.1 |
| | BTAD | 82.7/87.3 | 67.1/85.6 | 80.7/65.4 | 88.9/95.2 | 80.9/83.1 | 87.2/93.9 | **93.6/97.1** | 85.5/95.8 | 92.3/- | 91.1/96.4 |
| | MVTec3D | 58.7/90.4 | 51.7/89.4 | 62.5/78.6 | 59.6/94.3 | 58.8/83.4 | 59.5/96.4 | 64.5/95.4 | 74.1/96.8 | 68.9/- | **78.5/97.5** |
| | **Average** | 65.9/82.4 | 59.6/84.5 | 72.4/68.4 | 74.2/93.7 | 69.8/83.0 | 74.4/94.2 | 80.9/95.8 | 83.7/95.3 | 85.3/- | **87.1/96.2** |
| | Medical AD Dataset | | | | | | | | | | |
| | BraTS | 49.8/66.7 | 59.5/88.5 | 58.0/92.8 | 49.4/90.2 | 58.2/93.5 | 54.6/81.4 | 65.7/91.2 | 55.9/91.5 | **74.6**/- | 67.9/**94.3** |
| | Video AD Dataset | | | | | | | | | | |
| | ShanghaiTech | 56.2/77.6 | 55.9/79.4 | 73.8/87.0 | 70.4/85.6 | 71.8/87.8 | 72.7/87.3 | 78.4/88.5 | 78.5/88.1 | 68.7/- | **82.4/91.9** |
| | **All Average** | 61.6/79.0 | 58.9/84.3 | 70.2/75.6 | 69.4/91.8 | 68.2/85.6 | 70.8/90.9 | 78.0/93.8 | 78.2/93.5 | 80.8/- | **83.1/95.2** |
| **4-shot** | | | | | | | | | | | |
| | Industrial AD Datasets | | | | | | | | | | |
| | MVTecAD | 65.9/71.9 | 67.4/81.1 | 75.5/64.0 | 82.5/94.9 | 80.6/90.2 | 84.8/94.5 | 90.5/95.7 | **94.6**/94.2 | 94.5/- | 94.2/**96.9** |
| | VisA | 56.4/79.9 | 52.1/81.8 | 75.0/65.4 | 75.3/93.3 | 71.7/87.1 | 78.0/93.5 | 86.2/97.4 | 84.1/95.2 | 87.7/- | **90.8/97.5** |
| | BTAD | 82.7/87.3 | 67.1/85.6 | 81.7/65.5 | 89.9/95.8 | 84.0/89.4 | 90.8/94.9 | **95.6/97.6** | 87.2/95.8 | 91.7/- | 91.5/96.8 |
| | MVTec3D | 58.7/90.4 | 51.7/89.4 | 62.3/78.6 | 62.8/94.5 | 61.5/87.1 | 62.3/96.7 | 70.9/97.3 | 76.0/97.0 | 69.1/- | **82.4/97.9** |
| | **Average** | 65.9/82.4 | 59.6/84.5 | 73.6/68.4 | 77.6/94.6 | 74.5/88.5 | 79.0/94.9 | 85.8/97.0 | 85.5/95.6 | 85.8/- | **89.7/97.3** |
| | Medical AD Dataset | | | | | | | | | | |
| | BraTS | 49.8/66.7 | 59.5/88.5 | 66.3/94.8 | 60.6/94.5 | 71.2/95.9 | 60.0/87.3 | 74.7/94.0 | 67.3/93.2 | 76.9/- | **84.6/96.1** |
| | Video AD Dataset | | | | | | | | | | |
| | ShanghaiTech | 56.2/77.6 | 55.9/79.4 | 77.1/87.4 | 74.3/85.9 | 77.8/88.2 | 76.4/87.7 | 79.8/89.5 | 79.6/88.6 | 69.2/- | **84.3/92.6** |
| | **All Average** | 61.6/79.0 | 58.9/84.3 | 73.0/76.0 | 74.2/93.2 | 74.5/89.7 | 75.4/92.4 | 83.0/95.3 | 81.5/94.0 | 81.5/- | **88.0/96.3** |

**Competing Methods.** We select the representative one-for-one AD method (RDAD [13]) and the one-for-many AD method (UniAD [47]) as baselines. Our method is mainly compared with few-shot AD methods. Following WinCLIP [19], we adapt three conventional full-shot AD methods, including SPADE [11], PaDiM [12], and PatchCore [30], to the few-shot setting by making use of few-shot normal samples to calculate distance-based anomaly scores. We also compare with the few-shot AD method RegAD [18]. Most of these methods are based on WideResNet50 to extract features. However, these methods still need to re-model in new classes based on few-shot normal samples (see Sec.2), while our ResAD can be directly applied to new classes only requiring extracting features of few-shot normal samples as reference. Then, we also compare with the recent CLIP-based few-shot AD methods, including WinCLIP [19][5] and InCTRL [50]. To guarantee the rationality of result comparison, we ensure all methods use the same few-shot normal samples, and all results are evaluated based on $224 \times 224$ resolution.

## 4.2 Main Results

Tab.1 represents the comparison results of our ResAD and other SOTA competing methods in image-level AUROC and pixel-level AUROC, respectively, on six real-world AD datasets. Note that all

---

[5]No official implementation of WinCLIP is available. We use the public implementation at `https://github.com/zqhang/Accurate-WinCLIP-pytorch`.

the results are dataset-level average results across their respective data subsets. Compared to the results on known classes (results in the original papers), the performance of conventional AD methods will drop dramatically when used for new classes, whether it is the one-for-one[6] (RDAD) or the one-for-many (UniAD) AD method.

By comparison, we can see that our ResAD can significantly outperform all non-CLIP-based AD methods on both the 2-shot and 4-shot settings. With more few-shot normal images, the performance of all methods generally becomes better. On average, our ResAD outperforms the best competing model, RegAD, with up to 7.2%/2.9% and 7.6%/2.9% improvements under the 2-shot and 4-shot settings, respectively. In addition, please note that when evaluating RegAD, we utilize the few-shot normal samples to re-model the Multivariate Gaussian distribution for each new class (see Sec.2), while our ResAD is directly applied to each new class without any re-modeling or fine-tuning. Even with re-modeling, our method still has advantages over the conventional few-shot AD methods in cross-class detection.

We further implement a ResAD$^{\dagger}$ model by utilizing the powerful ImageBind [16] as the feature extractor. The outputs from the [8, 16, 24, 32] layers of ImageBind-Huge are used as the pre-trained features. ImageBind is a recently proposed large-scale pre-trained multimodal model, which shows emergent zero-shot and few-shot recognition capabilities across many vision tasks. As shown in Tab.1, by employing a model with stronger representation capability, our method can achieve better cross-dataset performance, which significantly outperforms the SOTA CLIP-based AD methods, WinCLIP and InCTRL. This demonstrates that our framework can effectively combine the latest vision models to manifest stronger class-generalizable ability. What's more, under the 4-shot setting, our ResAD by only using WideResNet50 can achieve comparable or even better results than WinCLIP and InCTRL (with more powerful CLIP-based ViT-B/16+), further demonstrating our superiority. Moreover, these two CLIP-based methods also heavily rely on CLIP-based image encoders. When we employ WideResNet50 in these two methods, our method has more advantages than these two methods (please see Appendix Tab.7).

When applied to other domains (medical images and video scenarios), our method also has better cross-domain generalization ability, despite it being trained on industrial data (the MVTecAD dataset).

### 4.3 Ablation Studies

In ablation studies, we conduct experiments under the "VisA to MVTecAD" case and use the commonly used WideResNet50 [49] as the feature extractor.

**Residual Feature Learning.** As shown in Tab.2(a), without residual feature learning, the cross-dataset performance drops dramatically from 90.5%/95.7% to 72.8%/82.9%. This verifies our confirmation that residual feature learning is of vital significance for class-generalizable anomaly detection. Analogously, any method that can reduce the variations of new class distribution relative to known class distributions is also promising to achieve class-generalizable anomaly detection.

**Feature Constraintor.** The ablation study on the effectiveness of the Feature Constraintor is also in Tab.2(a). "w/o Feature Constraintor" means the $\mathcal{L}_{occ}$ in E.q.(6) is not used. The effectiveness indicates that by further reducing the variations in the feature distribution and making the distribution of new classes more consistent with the learned distribution, we can achieve better cross-class AD results. In Fig.3, we also present a visualization figure to intuitively show the effect of the Feature Constraintor.

**Abnormal Invariant OCC Loss.** The effectiveness of abnormal invariant OCC loss is validated in Tab.2(a). "w/o Abnormal Invariant OCC Loss" means the $\mathcal{L}_{occ}$ only has the first part of E.q.(2). With the abnormal invariant OCC loss, image-level and pixel-level AUROCs can be improved by 5.6% and 1.8%, respectively. Moreover, we also find that without this loss, the results would rapidly decrease after certain epochs of training (*i.e.*, overfitting). This shows that keeping abnormal residual features as invariant as possible is beneficial to avoid the Feature Constraintor overfitting and thus achieve better results.

**Feature Constraintor Configuration.** We further ablate the network architectures of the Feature Constraintor, the results are shown in Tab.2(b). The results indicate that the simple Conv+BN+ReLU

---

[6]"one-for-one" means learning one specific AD model for each class, "one-for-many" means learning one AD model for multiple classes. However, they both don't consider new classes.

Table 2: Ablation studies on MVTecAD. (a) "Ours" implementation follows the same configuration as in Tab.1. "w/o..." indicates that we remove a certain component relative to "Ours". I-AUROC and P-AUROC mean image-level AUROC and pixel-level AUROC, respectively. (b) "ConvBnRelu" implements a simple Conv+BN+ReLU network. "BasicBlock" adopts the BasicBlock in ResNet. "BottleNeck" adopts the BottleNeck in ResNet. "MultiScaleFusion" is a FPN-like architecture to fuse multi-scale features. In "MultiScaleFusion+BasicBlock/BottleNeck", we add BasicBlock/BottleNeck after the multi-scale fusion.

(a) Framework ablation studies.

| Model | ‖ I-AUROC | P-AUROC |
|---|---|---|
| Ours | ‖ 90.5 | 95.7 |
| w/o Residual Feature Learning | ‖ 72.8 | 82.9 |
| w/o Feature Constraintor | ‖ 82.3 | 93.5 |
| w/o Abnormal Invariant OCC Loss | ‖ 84.9 | 93.9 |

(b) Comparison of different feature constraintors.

| Network Architecture | ‖ I-AUROC | P-AUROC |
|---|---|---|
| ConvBnRelu | ‖ 90.5 | 95.7 |
| BasicBlock | ‖ 87.6 | 94.4 |
| BottleNeck | ‖ 86.0 | 94.4 |
| MultiScaleFusion | ‖ 84.1 | 92.9 |
| MultiScaleFusion+BasicBlock | ‖ 82.3 | 92.9 |
| MultiScaleFusion+BottleNeck | ‖ 81.0 | 93.3 |

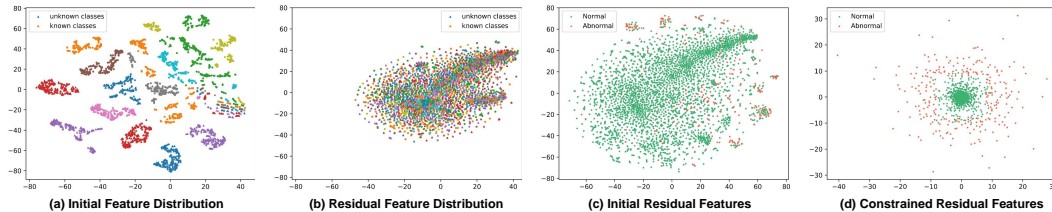

(a) Initial Feature Distribution    (b) Residual Feature Distribution    (c) Initial Residual Features    (d) Constrained Residual Features

Figure 3: Feature t-SNE visualization. (a) In the initial feature space, the features from different classes are significantly different. (b) In the residual feature space, even the residual feature distribution of unknown classes would not remarkably shift from the known distribution. Note that in (a) and (b), we only show normal residual features and use different colors to represent different classes. (c) The initial residual features. (d) The residual features after the Feature Constraintor.

network can yield the best performance. We observe a significant performance drop with a more complex feature constraintor (*e.g.*, Bottleneck, MultiScaleFusion). One possible reason is that a complex network may lead to overfitting, reducing the generalization ability for various anomalies in new classes.

**Cross-Class Within One Dataset.** We show the results of training with $n$ classes from MVTecAD and testing on the remaining $15-n$ classes. By varying $n$, we can demonstrate the sensitivity of the model to different numbers of training classes. Note that different $n$ means the number of test classes is different (this will cause the test results of different $n$ cannot be compared with each other). Thus, we use fixed 5 classes as the test classes, including hazelnut, pill, tile, carpet, and zipper. For $n = 5$, the training classes include bottle, cable, capsule, grid, and leather. For $n = 10$, the training classes include bottle, cable, capsule, grid, leather, metal nut, screw, toothbrush, transistor, and wood. The results under the 4-shot setting are in Tab.3. The results demonstrate that cross-dataset generalization is more challenging than cross-class generalization in a single dataset. With more training classes, the results will be better, but the model is not very sensitive.

Table 3: Cross-class results with different numbers of training classes n.

| $n = 5$ | $n = 10$ | VisA to MVTecAD |
|---|---|---|
| 96.4/97.6 | 96.8/97.9 | 95.1/97.2 |

### 4.4 Generalization to Other Anomaly Detection Frameworks

Furthermore, we think that our residual feature learning insight is not limited to the model proposed in this paper, but can be considered as an effective and general method for solving class-generalizable anomaly detection. The main reasons are: 1) The process of converting initial features to residual features can be easily applied to other AD models. 2) Residual features are less sensitive to new classes (see Sec.3.1). In this subsection, we further extend our method to the popular reconstruction-based AD framework. Specifically, we employ UniAD [47] as baseline and incorporate our method into it. As UniAD is feature-based AD method, combining our residual feature learning with it is straightforward.

Table 4: Anomaly detection and localization results when incorporating our method into UniAD. "RFL" represents residual feature learning.

| | MVTecAD | VisA | BTAD | MVTec3D |
|---|---|---|---|---|
| **UniAD** [47] | 67.4/81.1 | 52.1/81.8 | 67.1/85.6 | 51.7/89.4 |
| + RFL (Ours) | 93.0/94.9 | 72.7/86.1 | 87.3/94.0 | 76.7/96.9 |
| Δ | +25.6/13.8 | +20.4/3.3 | +20.0/8.4 | +25.0/7.0 |

We can convert the initial features into residual features and then perform subsequent feature reconstruction. The experimental results are shown in Tab.4. It can be found that the performance of UniAD is quite poor when used for new classes, while converting to residual feature learning can significantly improve the model's class-generalizable capacity. The remarkable improvements (*e.g.*, 25.6%/13.8% on MVTecAD) validate the effectiveness and generalizability of residual features for designing generalizable AD models.

## 4.5  Visualization and Qualitative Results

**Visualization Results.** Fig.3(a) and (b) show the t-SNE visualization of initial features and residual features. It can be found that in the initial feature space, the feature distribution of new classes is significantly different from the distribution of known classes, resulting in poor adaptability of AD models to new classes. However, the variations between different classes can be significantly reduced by converting into residual space. In this way, the model's generalizability to new classes can be effectively improved. Fig.3(c) and (d) show the t-SNE visualization of initial residual features and residual features after the Feature Constraintor. Results show that the Feature Constraintor can make the normal residual features more compact and more separated from the abnormal features.

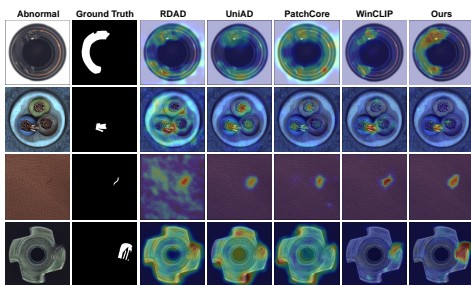

Figure 4: Qualitative results. The anomaly score maps are generated under the "VisA to MVTecAD" case.

**Qualitative Results.** Fig.4 shows qualitative results under the "VisA to MVTecAD" case with WideResNet50 as the feature extractor. It can be seen that most SOTA methods fail to generate good anomaly localization maps for new classes, mainly existing many false positives in normal regions. However, our method can effectively avoid false positives in normal regions and locate anomalies more accurately. More qualitative results are in Appendix Fig.5.

## 5  Conclusion

In this paper, we propose a simple but effective framework: ResAD, for achieving class-generalizable anomaly detection. ResAD consists of several simple neural network modules that are easy to train and apply in real-world scenarios. Despite the simplicity, ResAD achieves remarkable anomaly detection results in new classes. We conclude our findings for future research: residual features are really effective for designing generalizable AD models, and our feature constraining insight also has good reference values for future work.

**Limitations.** The limitations of our method are discussed in Appendix C.

**Social Impacts.** As a unified model for class-generalizable anomaly detection, the proposed method does not suffer from particular ethical concerns or negative social impacts. All datasets used are public. All qualitative visualizations are based on industrial product images, which does not infringe personal privacy.

## Acknowledgments

This work was supported in part by the National Natural Science Fund of China (62371295), the Shanghai Municipal Science and Technology Major Project (2021SHZDZX0102), and the Science and Technology Commission of Shanghai Municipality (22DZ2229005).

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

# Appendix

## A    More Discussions

### A.1    Detailed Comparison with InCTRL

We should have proposed the idea of residual learning independently of InCTRL and almost at the same time (*i.e.*, we completed the initial version of our method during CVPR 2024). But our method has obvious differences with InCTRL in the definition and utilization of residuals. This (*i.e.*, two independent works almost simultaneously proposed the residual learning idea) also demonstrates residual learning is an effective way to achieve class-generalizable anomaly detection. The main differences between our method and InCTRL [50] are as follows:

(1) The definition of residuals in InCTRL is based on feature distances. The residual map is defined by ((E.q.(1) in the InCTRL paper): $M_x^l(i, j) = 1 - \langle T_x^l(i, j), h(T_x^l(i, j)|\mathcal{P}') \rangle$, where $h(T_x^l(i, j)|\mathcal{P}')$ returns the embedding of the patch token that is most similar to $T_x^l(i, j)$ among all image patches in $\mathcal{P}'$, and $\langle \cdot \rangle$ is the cosine similarity function. Thus, InCTRL is based on residual distance maps, while our method is based on residual features.

By comparison, we think that residual distances in InCTRL can limit the range of residual representation (as the cosine similarity is in [-1,1]). This is not beneficial for distinguishing between normal and abnormal regions, as a position on the residual map is only represented by a residual distance value. Within a limited representation range ($1-[-1,1] \rightarrow [0,2]$), normal and abnormal residual distance values are more likely to be not strictly separable. Thus, for a position on the residual map, it's hard for us to make decision based on a scalar value. So, InCTRL makes image-level classification based on a whole residual map (see the following (2)). In contrast, our residual features don't limit the range of residual representation and can retain the feature properties. In high-dimensional feature space, we can also establish better decision boundaries between normal and abnormal (a basic idea in machine learning: solving low-dimensional inseparability by converting to high-dimension).

(2) InCTRL devises a holistic anomaly scoring function $\phi$ to learn the residual distance map $M_x = \frac{1}{n} \sum_{l=1}^{n} M_x^l$ and convert it to an anomaly score: $s(x) = \phi(M_x^+; \Theta_\phi) + \alpha s_p(x)$ (E.q.(8) in the InCTRL paper), where $M_x^+ = M_x \oplus s_i(x) \oplus s_a(x)$ (E.q.(7)). $s_i(x)$ is an anomaly score based on an image-level residual map $F_x$ (see E.q.(4) in the InCTRL paper) and $s_a(x)$ is a text prompt-based anomaly score. Thus, InCTRL is to train a binary classification network based on residual distance maps. For each input image, InCTRL finally only outputs an image-level anomaly score. Our method is to learn the distribution of residual features, an anomaly score can be estimated for each feature, thus can be used to locate anomalies.

(3) Due to the designs in InCTRL that we mentioned above, one main advantage of our method is that it can achieve image-level anomaly detection and also pixel-level anomaly localization, while InCTRL only achieves image-level anomaly detection functionality.

As for performance, the average results on six AD datasets of our method are better than InCTRL's (please see Tab.1 in the main text).

### A.2    Discussion on Few-Shot Normal Sample Selection Strategy

In our paper, the few-shot normal reference samples are randomly selected and fixed. This will raise concerns about whether random selection is reasonable and whether it may lead to insufficient representativeness of the reference feature pools. From the perspective of method comparison, we think that random selection is feasible, as long as we ensure that all methods use the same reference samples, the result comparison is reasonable. However, when the difference between normal images is too large, it may cause the reference feature pools are not representative. Nonetheless, please further note that, in our method, we only extract the features of the few-shot normal samples and store all the features in the reference feature pools. The reference feature pools don't impair or lose any representation features. For the few-shot normal samples, the reference feature pools are representative enough to them. Therefore, whether the representativeness is sufficient is determined by the few-shot normal samples themselves rather than our method. For some classes, the few-shot normal samples are representative, while for some hard classes, they may mot be representative enough.

For practical applications, this issue should be particularly focused and reasonably addressed. We expect that the reference samples can fully represent their class, so it's best to have sufficient differences between the reference samples. Thus, the sample selection strategy cannot be random. Of course, the simplest resolution is to increase the number of reference samples. This is feasible, as in practical applications, the number of reference samples is usually not as strict as the 2-shot and 4-shot in our paper.

A feasible method is to first cluster all available normal samples into different clusters based on a clustering algorithm (*e.g.*, KMeans). Then, based on the number of reference samples, we evenly distribute it to each cluster. When selecting from a cluster, we can prioritize selecting samples closer to the center. During clustering, we think that the FID and LPIPS metrics are good ways to calculate the difference between two samples. In addition, when there are a large number of reference samples, we can also use the method in PatchCore [30] to select coreset features as reference features, which will be more efficient and also representative.

### A.3    Feature Constraintor and Feature Distribution Estimator

The goal of our Feature Constraintor is to constrain initial residual features to a spatial hypersphere for further reducing feature variations. After the Feature Constraintor, feature variations can effectively be reduced, but this does not mean that the feature distribution is fixed within the hypersphere. The ideal situation is that even in new classes, normal feature distribution is fixed within a hypersphere, while all anomalous features are outside the hypersphere. Then, only the Feature Constraintor part is enough to achieve good AD results. However, in practical optimization, it's hard to achieve the ideal situation. After the Feature Constraintor, normal and abnormal features may still not be fully separable based on the distances from the features to the center. Therefore, the Feature Distribution Estimator (namely the normalizing flow model used in our method) is used to learn the feature distribution, which can assist us in better distinguishing normal and abnormal features.

## B    Model Architecture and Complexity

**Normalizing Flow Model Architecture**. The normalizing flow model is mainly based on Real-NVP [14] architecture, which is composed of the so-called coupling layers. All coupling layers have the same architecture, where a learnable subnet is utilized to predict the affine parameters [14]. The convolutional subnet in Real-NVP is replaced with a two-layer MLP network. Each coupling layer is followed by a random and fixed soft permutation of channels [2] and a fixed scaling by a constant, similar to ActNorm layers introduced by [20]. Furthermore, we adopt the soft clamping of multiplication coefficients used by [14]. Following [17], we add positional embeddings to each coupling layer, which are concatenated with the first half of the input features. The dimension of all positional embeddings is set to 256.

**Complexity Comparison.** With the image size fixed as $224 \times 224$, we compare the number of parameters and per-image inference time with all competitors. We conclude that the advantage of ResAD does not come from a larger model capacity.

Table 5: Complexity comparison between our ResAD and other competing methods.

|  | RDAD | UniAD | SPADE | PaDiM | PatchCore | RegAD | WinCLIP | InCTRL | ResAD | ResAD$^{\dagger}$ |
|---|---|---|---|---|---|---|---|---|---|---|
| Parameters(M) | 150.6 | 6.3 | 74.5 | 686.9 | 69.5 | 25.2 | 165.9 | 117.5 | 59.2 | 442.6 |
| Infer time(fps) | 5.6 | 24.4 | 4.8 | 14.1 | 21.5 | 20.2 | 0.51 | 0.53 | 21.3 | 18.8 |

## C    Limitations

In this paper, we propose a simple but effective AD framework, ResAD, to accomplish class-generalizable anomaly detection. Even if our method manifests good AD performance on six real-world industrial AD datasets, there are still some limitations of our work. One limitation of our work is that we only conducted experiments on data of image modality, it's very valuable to further extend our method to other application domains and data modalities, such as video data and time series, to more comprehensively validate our method's generalizability. Our future work will focus

on further generalizing our method to other data modalities, not only to achieve class-generalizable but also domain-generalizable anomaly detection. Another valuable future work is to incorporate our method into recent SOTA AD methods for achieving better class-generalizable AD performance. In Sec.4.4, we incorporate our method into UniAD and gain remarkable improvements. How to upgrade the other types of anomaly detection methods to class-generalizable AD methods and how to find a general approach for class-generalizable (or even domain-generalizable) anomaly detection will be the future works.

## D    Datasets

**MVTecAD.** The MVTecAD [5] dataset is widely used as a standard benchmark for evaluating unsupervised anomaly detection methods. This dataset contains 5354 high-resolution images (3629 images for training and 1725 images for testing) of 15 different product categories. 5 classes consist of textures and the other 10 classes contain objects. A total of 73 different defect types are presented and almost 1900 defective regions are manually annotated in this dataset.

**BTAD.** The BeanTech Anomaly Detection dataset [25] is an another popular benchmark, which contains 2830 real-world images of 3 industrial products. Product 1, 2, and 3 of this dataset contain 400, 1000, and 399 training images respectively.

**MVTec3D.** The MVTec3D [7] dataset is for 3D anomaly detection, which contains 4147 high-resolution 3D point cloud scans paired with 2D RGB images from 10 real-world categories. In this dataset, most anomalies can also be detected only through RGB images. Since we focus on image anomaly detection, we only use RGB images of the MVTec3D dataset.

**VisA.** The Visual Anomaly dataset [51] is a larger anomaly detection dataset compared to MVTecAD [5]. This dataset contains 10821 images with 9621 normal and 1200 anomalous samples. In addition to images that only contain single instance, the VisA dataset also have images that contain multiple instances. Moreover, some product categories of the VisA dataset, such as Cashew, Chewing gum, Fryum and Pipe fryum, have objects that are roughly aligned. These characteristics make the VisA dataset more challenging than the MVTecAD dataset, whose images only have single instance and are better aligned.

## E    Sensitivity of Balancing The Loss Terms

During training, we found that summing up the three loss terms (see E.q.(6)) and then backpropagating gradients to optimize the whole model would lead to unstable training. Then, we used the "*torch.detach()*" method in the Pytorch library to detach the features after the Feature Constraintor and then sent the detached features into the normalizing flow (NF) model. This simple way can make the model training more stable. Thus, the weight of $\mathcal{L}_{occ}$ can be set as 1 (*i.e.*, we can not need to balance $\mathcal{L}_{occ}$ with $\mathcal{L}_{ml}$ and $\mathcal{L}_{bg-spp}$, as the Feature Constraintor and the NF model parts are separated in the gradient graph). When training the NF model, the $\mathcal{L}_{ml}$ is the basic loss. Thus, we keep the weight of $\mathcal{L}_{ml}$ as 1 and set a variable $\lambda$ as the weight of $\mathcal{L}_{bg-spp}$. By varying different $\lambda$ values, the results (under the 4-shot setting, from VisA to MVTecAD) about the sensitivity are in Tab.6.

Table 6: Anomaly detection and localization results when varying different $\lambda$ values for balancing the three loss terms.

| $\lambda$ | 0.1 | 0.5 | 1 | 2 | 3 | 5 | 10 |
|---|---|---|---|---|---|---|---|
| | 89.1/94.9 | 90.0/95.6 | 90.5/95.7 | 90.6/96.0 | 89.8/95.3 | 89.3/95.2 | 88.7/94.9 |

Both small and large $\lambda$ values can lead to performance degradation. $\mathcal{L}_{bp-spp}$ is to assist model in learning abnormal residual features. Small $\lambda$ may cause the impact of abnormal features on the whole loss $\mathcal{L}$ (E.q.(6)) to be relatively small. Large $\lambda$ may lead to overfitting to known anomalies, which is not conducive to generalization.

# F   Additional Results

**Extra Few-shot AD Results.** In the main text, the WinCLIP and InCTRL utilize the CLIP-based ViT-B/16+ as the feature extractor. We further employ the WideReset50, which is commonly used in anomaly detection, as the feature extractor in these methods. We note that the two methods do not necessarily need CLIP-based vision encoders. We can remove the vision-language alignment part in the two methods and the remaining modules can also achieve anomaly detection. For example, we can send image patches provided by WinCLIP's window mechanism into WideResNet50, and also obtain the window embedding maps of different scales as shown in Figure 4 of the WinCLIP paper. However, because the features of WideResNet50 are not aligned with the text features, we remove the language-guided anomaly score map and only generate the vision-based anomaly score map based on the few-shot normal samples (the WinCLIP+ in the WinCLIP paper). The results under the 4-shot setting are in Tab.7. The results show that our method has more significant superiorities on networks with weaker representation capability. Thus, compared to WinCLIP and InCTRL, our method is less reliant on the representation capability of the backbone network and is more widespreadly applicable for various backbones.

Table 7: Anomaly detection and localization results with WideResNet50 as the feature extractor.

| Dataset | WinCLIP | InCTRL | ResAD |
|---|---|---|---|
| **MVTecAD** | 86.6/91.6 | 86.9/- | 90.5/95.7 |
| **VisA** | 80.7/92.5 | 82.3/- | 86.2/97.4 |
| **BTAD** | 87.7/93.7 | 90.4/- | 95.6/97.6 |
| **MVTec3D** | 63.1/91.7 | 63.2/- | 70.9/97.3 |

**Additional Results on Other Data Groups.** In the main text, the results are evaluated on a single group of few-shot reference samples. However, the selection of few-shot reference samples may affect the performance of the model. To fully represent our model's robustness, we further randomly select two groups of few-shot reference samples. The results under the 4-shot setting are in Tab.8.

Table 8: Anomaly detection and localization results on other two groups of few-shot reference samples.

| Dataset | Group1 | Group2 | Results in Tab.1 | Mean±Std |
|---|---|---|---|---|
| **MVTecAD** | 91.0/96.0 | 90.7/95.9 | 90.5/95.7 | 90.7±0.21/95.9±0.12 |
| **VisA** | 86.3/97.5 | 86.9/97.6 | 86.2/97.4 | 86.5±0.31/97.5±0.08 |
| **BTAD** | 95.3/97.5 | 95.4/97.6 | 95.6/97.6 | 95.4±0.12/97.6±0.05 |
| **MVTec3D** | 70.2/97.1 | 70.5/97.3 | 70.9/97.3 | 70.5±0.29/97.2±0.09 |

**Additional Qualitative Results.** We present in Fig.5 additional anomaly localization results of categories from the MVTecAD dataset. The anomaly score maps are generated under the "VisA to MVTecAD" case, where AD models are trained on the VisA dataset.

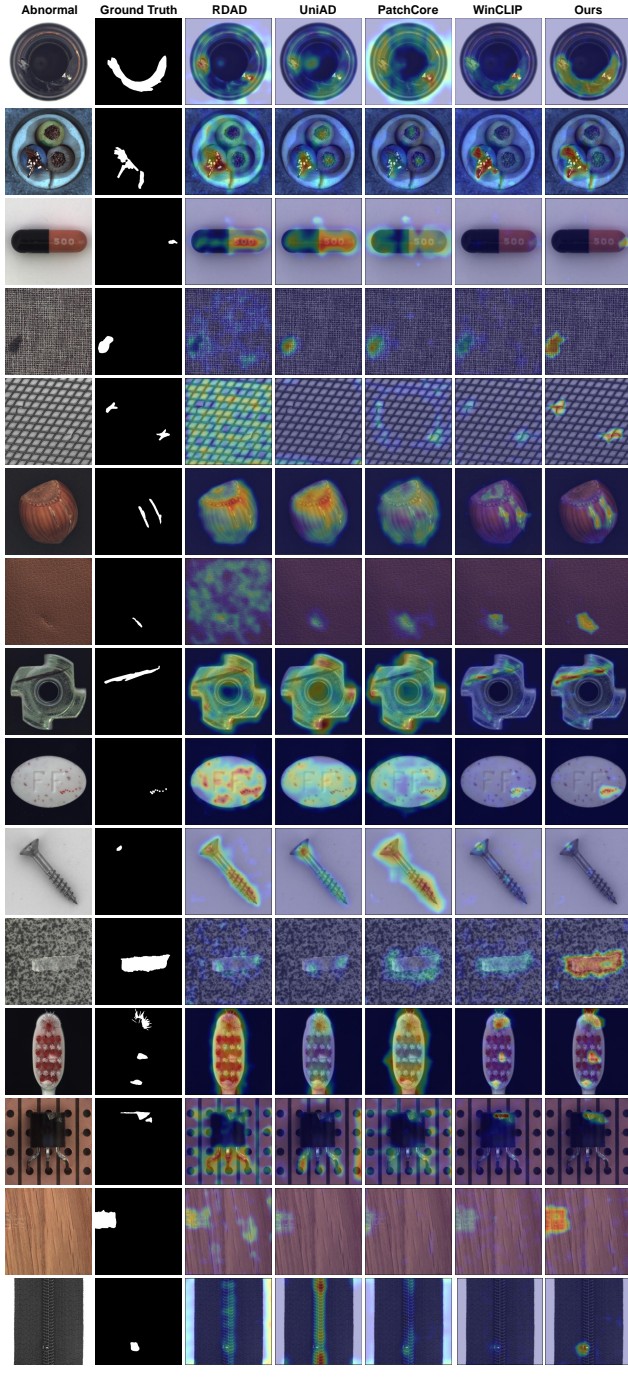

Figure 5: Additional qualitative results on MVTecAD.

