# OpenReview forum: "ResAD: A Simple Framework for Class Generalizable Anomaly Detection"
_NeurIPS.cc/2024/Conference — NeurIPS 2024 spotlight_

### Official Review · Reviewer_6bzj · 2024-06-24

**Soundness:** 3
**Presentation:** 3
**Contribution:** 2
**Rating:** 6
**Confidence:** 5

**Summary:**

The paper analyzes the class-generalizable anomaly detection problem and introduces residual feature learning.
Based on the residual features, the paper proposes a simple AD framework, i.e., ResAD, which incorporates OCC loss and distribution estimating to distinguish normal and abnormal data.
The experimental results demonstrate that the ResAD performs well on real-world industrial AD datasets.

**Strengths:**

1. The paper analyzes the few-shot class generalizable anomaly detection problem and delivers an interesting insight into residual features.
2. The proposed method is intuitive and easy to understand.
2. The paper is well-written and organized.

**Weaknesses:**

1. The residual learning for few-shot AD has already been proposed in inCTRL[1]. The proposed Multi-Layer Patch-Level Residual Learning scheme in InCRTL is more sophisticated and reasonable than the direct subtraction in this paper.
2. The results in Table 1 of InCTRL are not consistent with the results in the original paper. Compared with the original results of InCTRL, the ResAD results do not achieve the SOTA performance.
3. The paper aims to achieve generalization across different classes. I think the authors should compare the accuracy of each class on the Visa dataset with other methods to demonstrate the generalization capability of your approach for different classes, rather than taking the average accuracy of different classes in the dataset.

[1]Jiawen Zhu and Guansong Pang. Toward generalist anomaly detection via in-context residual learning with few-shot sample prompts. In CVPR, 2024.

**Questions:**

1. What is the superiority of the proposed simple subtraction-based residual learning compared with the residual learning in InCTRL?
2. In the related work, the author claims that the CLIP-based methods are difficult to generalize to anomalies in diverse classes. However, according to the experiment results, the proposed methods only perform well on industrial AD datasets while InCTRL performs well on various types of datasets, including Medical datasets and  Semantic datasets. Why does your method only compare different classes on industrial datasets, instead of comparing against anomaly datasets from other domain?  I am wondering how the ResAD performs on the datasets from other domains.
3. What are the main advantages of ResAD compared with WinCLIP and InCTRL since the generalization ability and complexity of ResAD are not as good as WinCLIP and InCTRL.
4. In the residual feature construction process, the residual feature is highly related to the closest normal reference features in the reference feature pool. Are the few-shot reference samples enough to represent the class-related attributes?
5. In table 1, the authors mention that RDAD and UniAD don't utilize the few-shot normal samples to fine-tune, so the results under 2-shot and 4-shot are the same.  RDAD and UniAD don’t require few-shot normal samples to fine-tune or refer, while the proposed method provide few normal samples to refer, so I believe it is meaningful to compare your method with those that require few-shot normal samples to refer, such as inctrl and winclip. Comparing it with RDAD and UniAD seems to be unfair especially in Table 3 . How do the results of corporating proposed method into WinCLIP and InCTRL?

**Limitations:**

The authors did not give a discussion on the limitations of the proposed method.

---

> ### Author Rebuttal · Authors · 2024-08-05
>
> **[To W1 and Q1].** Thanks for your professional review. We think that our work and InCTRL should be concurrent work. We also initially submitted our work to CVPR2024, and received two weak accepts and one reject (you can see the relevant materials in the rebuttal pdf file). Regretfully, our work was rejected at that time. So, we should have proposed the idea of residual learning independently of InCTRL and almost at the same time. But our method has obvious differences with InCTRL in the definition and utilization of residuals. In CVPR2024, our paper was rejected mainly because a reviewer thought our result comparison with unsupervised AD methods was unreasonable and unfair. In this submission, we mainly compare with few-shot AD methods and InCTRL. **Please see our response to Reviewer fL7E's Weakness 1, we provide a detailed comparison between our method and InCTRL**. Based on the comparison, we think that our method has the following advantages:
>
> (1) One main advantage of our method is that it can achieve image-level anomaly detection and also pixel-level anomaly localization, while InCTRL only achieves image-level anomaly detection (due to the designs in their method).
>
> (2) Compared to residual distance maps (in InCTRL), residual features are easier to integrate into other feature-based AD methods. For example, in Sec.4.4, we combine the residual feature learning with UniAD and RDAD, which can effectively improve the models’ class-generalizable capacity. In InCTRL, the authors devise an anomaly scoring network to learn the residual distance maps. The residual distance maps seem not easy to integrate into other AD methods (based on our analysis in the response to Reviewer fL7E’s Weakness 1).
>
> **[To W2].** We checked the results again and found that the results of InCTRL we reported were the same as in its original paper (94.0 and 85.5 under the 2-shot setting, 94.5 and 87.7 under the 4-shot setting). In the InCTRL paper, there are two tables, the AUROC results are in Table 1, and Table 2 are the AUPRC results, which are overall higher than the AUROC results. The original results you mentioned may be from Table 2 of the InCTRL paper. In Table 1 of our paper, we report the AUROC results, the average results on four AD datasets of our method are also better than InCTRL’s.
>
> **[To W3].** Thanks for your suggestion. Due to the page limitation and the need to extensively validate the effectiveness of our method on multiple datasets, we adopted to report the dataset-level average results, which was also the way adopted in the InCTRL paper. We present the detailed results on the VisA dataset in the rebuttal pdf file and also the following Comment.
>
> **[To Q2].** When writing the paper, we thought the results on these four industrial AD datasets were enough to validate the effectiveness of our method (In Table 1, our method can achieve better average results on these datasets than other methods), so we didn't consider more datasets from other domains. Please see our response to Weakness 1 of Reviewer CQaQ, we further run experiments on a medical dataset and a video AD dataset.
>
> **[To Q3].** According to our response to Weakness 2, our method is better than InCTRL and WinCLIP in terms of the average results on four AD datasets. The advantages to InCTRL are discussed in **[To W1 and Q1]**. Compared to WinCLIP, we think the advantages are:
>
> (1) WinCLIP generates anomaly score maps by directly calculating the similarity between vision and text features. The model is not trained on AD datasets. Thus, WinCLIP is more likely to rely on the visual-language comprehension abilities of CLIP. When text prompts cannot capture the desired anomaly semantics, the results may be poor. In addition, WinCLIP requires text prompts, which can bring extra complexity, while our method does not require any text.
>
> (2) Due to its sliding window mechanism, WinCLIP has low efficiency. In Table 4 in the Appendix, we provide the number of parameters and per-image inference time of our method and other methods (we list the table in the rebuttal pdf file and also the following Comment). The inference speed of WinCLIP is 0.51fps, while our method's is 18.8fps.
>
> **[To Q4].** Please see our response to Weakness 3 of Reviewer 226V.
>
> **[To Q5].** Yes, in Table 1, our method is mainly compared with these few-shot AD methods. Including the cross-dataset results of UniAD and RDAD is not for comparison, but to demonstrate that conventional one-for-one (RDAD) and also one-for-many (UniAD) AD methods cannot be directly applied to new classes (compared to the results in their original papers). Thus, achieving class-generalizable anomaly detection requires new insights and specific designs. In Table 3, we mainly aim to demonstrate that the residual feature learning can be easily incorporated into conventional feature-based AD methods and can effectively improve their class-generalizable capacity.
>
> We also considered combining our method with WinCLIP and InCTRL but found it not easy to achieve. In WinCLIP, the model is based on the alignment between vision and text features. As the semantics of residual features and initial features are different, converting to residual features can lead to misalignment with text features. In addition, our method is to learn the residual feature distribution, while WinCLIP doesn't have the training stage on AD datasets. In InCTRL, the model is designed based on residual distance maps. InCTRL devises an anomaly scoring network (discriminative model) to learn the residual distance map and convert it to an anomaly score. Our method is designed based on residual features, which utilizes a normalizing flow model (probabilistic generative model) to learn the residual feature distribution. The obvious differences in the definition and utilization of residuals between our method and InCTRL make integrating with InCTRL also not easy.
>
> If you still have any questions, we are very glad to further discuss with you.

---

> ### Author Response · Authors · 2024-08-05
>
> **[To W3].** Below, we present the detailed results on the VisA dataset under the 4-shot setting.
>
>  |   | RDAD | UniAD | SPADE | PaDiM | PatchCore  | RegAD  | ResAD  | WinCLIP | InCTRL | ResAD$^{†}$ |
> | --- | --- | --- | --- | --- | ---  | ---  | ---  | --- | --- | --- |
>  | Candle | 51.9/63.0 | 58.5/76.4 | 73.0/58.2 | 87.4/97.6 | 79.9/92.9 | 85.3/97.4 | 89.8/99.1 | **96.9**/96.0 | 93.7/- | 93.3/**99.3** |
>   | Capsules | 39.8/73.9 | 57.3/74.6 | 68.4/48.3 | 55.8/78.3 | 61.0/83.7 | 59.8/75.3 | 72.9/**98.0** | 83.5/96.1 | 85.9/- | **86.7**/97.9 |
>  | Cashew | 64.4/80.4 | 40.0/87.5 | 85.8/81.6 | 74.5/98.8 | 88.7/87.1 | 84.0/97.4 | 93.4/**98.2** | 94.7/96.8 | **97.8**/- | 95.7/98.0 |
>   | Chewinggum | 72.0/78.4 | 47.2/88.9 | 85.4/77.6 | 94.9/97.5 | 95.5/97.9  | 93.8/98.0 | 97.9/**99.4** | 97.8/99.0 | **99.8**/- | 97.7/98.9 |
>    | Fryum | 72.6/87.3 | 51.2/81.4 | 77.6/77.4 | 64.7/94.1 | 68.9/83.4  | 79.6/94.4 | 86.7/92.5 | 86.8/**95.2** | **96.7**/- | 94.7/92.8 |
>    | Macaroni1 | 54.1/86.9 | 48.8/85.5 | 65.8/62.4 | 62.8/92.1 | 62.7/76.6  | 71.0/94.9 | 88.3/**99.4** | 89.1/92.9 | 77.6/- | **91.4**/98.7 |
>    | Macaroni2 | 53.9/91.7 | 56.8/89.4 | 45.2/45.1 | 64.4/87.8 | 61.2/75.8  | 61.9/88.6 | **77.6**/**98.0** | 73.9/89.8 | 74.6/- | **77.6**/96.9|
>    | Pcb1 | 51.9/70.5 | 58.4/74.0 | 89.3/56.3 | 81.1/89.9 | 80.8/94.2  | 79.9/94.4 | 83.7/97.6 | 84.6/96.3 | **95.9**/- | 90.5/**98.8** |
>    | Pcb2 | 66.5/78.4 | 47.8/75.5 | 75.5/64.8 | 69.2/93.8 | 71.0/82.0  | 73.8/94.1 | 68.9/94.7 | 61.1/92.3 | 66.9/- | **85.3**/**96.9** |
>    | Pcb3 | 58.4/82.0 | 50.0/84.0 | 75.0/54.4 | 69.1/95.2 | 57.2/91.2  | 73.0/96.8 | 81.4/**96.7** | 72.1/94.6 | 76.1/- | **84.6**/95.7 |
>    |  Pcb4 | 21.6/74.4 | 62.5/72.5 | 86.8/68.5 | 91.6/94.8 | 47.3/83.2  | 82.0/92.3 | 95.0/96.3 | 76.2/96.9 | **97.5**/- | 93.9/**98.0**
>    | Pipe\_fryum | 70.2/92.0 | 46.7/91.4 | 72.0/90.4 | 88.7/99.2 | 85.9/97.2  | 92.1/98.9 | 98.9/**98.5** | 92.0/96.5 | 86.9/- | **99.3**/**98.5** |
>    | **Average** | 56.4/79.9 | 52.1/81.8 | 75.0/65.4 | 75.3/93.3 | 71.7/87.1 | 78.0/93.5 | 86.2/97.4 | 84.1/95.2 | 87.7/- | **90.8**/**97.5** |
>
> The detailed results show that our method can achieve better results in most classes. In the revision, we will add the above detailed results and also detailed results of other datasets to the Appendix.
>
> **[To Q3].** Below, we present the table of computation complexity.
>
>
>  | | RDAD | UniAD | SPADE  | PaDiM  | PatchCore | RegAD | WinCLIP | InCTRL | ResAD | ResAD$^{†}$ |
>  | --- | --- | --- | ---  | ---  | --- | --- | --- | --- | --- | --- |
>  | Parameters(M) | 150.6 | 6.3 | 74.5 | 686.9 | 69.5 | 25.2 | 165.9 | 117.5 | 59.2 | 116.4 |
>  | Infer time(fps) | 5.6 | 24.4 | 4.8 | 14.1 | 21.5 | 20.2 | 0.51 | 0.53 | 21.3 | 18.8 |
>
> **[Limitations].** In Appendix Sec.B, we provide discussions about our method's limitations. In the revision, we will further discuss the limitations of our method more comprehensively and clearly based on the comments of all reviewers.

---

> > ### Comment · Reviewer_6bzj · 2024-08-08
> >
> > Thanks for your response. Some responses have addressed part of my questions, but some issues still have not been well resolved.
> >
> > Regarding my Q4, based on the author's response that the reference feature pool is not representative, I think that this approach is not suitable for few-shot scenarios. Simply searching for the nearest nominal reference feature from the reference feature pool to achieve the learning goal of few-shot AD is insufficient.
> >
> > According to the results on the VisA dataset under the 4-shot setting and 2-shot setting and the table in the rebuttal for  Reviewer fL7E, the proposed method does not show significant advantages and sometimes even performs worse than other methods.

---

> > > ### Author Response · Authors · 2024-08-09
> > >
> > > We greatly appreciate your further response. We hope the following response can answer your question.
> > >
> > > **[To R1].** We respectfully argue that our response does not mention that the reference feature pool is not representative. We think the issue of representativeness you mentioned is not caused by our method, but caused by few-shot normal samples themselves. In our response, we state that ''when the difference between normal images is too large, it **may** cause the reference feature pool is not representative''. With this statement, we want to express that when the normal patterns of one product class are diverse and complex, the few-shot normal samples may lack some normal patterns (*e.g.*, if one class has five colors and one image only contains one color, then 4 reference samples can only contain a maximum of four colors), thus they are not enough to represent the class they belong to. In our method, we only extract the features of the few-shot normal samples and store all the features in the reference feature pool. The reference feature pool does not impair or lose any representation features. For the few-shot normal samples, it is representative enough to them. Therefore, whether the representativeness is sufficient is determined by the few-shot normal samples. For some classes, the few-shot normal samples are representative, while for some hard classes, they may mot be representative enough. For example, in our response to Weakness 3, the results of Macaroni2 and Pipe\_fryum classes are as follows:
> > >
> > > | | RDAD | UniAD | SPADE | PaDiM | PatchCore  | RegAD  | ResAD  | WinCLIP | InCTRL | ResAD$^{†}$ |
> > >  | --- | --- | --- |  --- | --- | --- | --- | --- | --- | --- | --- |
> > >  | Macaroni2 | 53.9/91.7 | 56.8/89.4 | 45.2/45.1 | 64.4/87.8 | 61.2/75.8 | 61.9/88.6 | **77.6**/**98.0** | 73.9/89.8 | 74.6/- | **77.6**/96.9 |
> > > | Pipe\_fryum | 70.2/92.0 | 46.7/91.4 | 72.0/90.4 | 88.7/99.2 | 85.9/97.2  | 92.1/98.9 | 98.9/**98.5** | 92.0/96.5 | 86.9/- | **99.3**/**98.5** |
> > >
> > > For the Pipe\_fryum class, the few-shot reference samples are representative enough. For the Macaroni2 class, the few-shot reference samples are not representative enough, and this issue exists for all AD methods that use few-shot normal samples, as it is limited by the data rather than the method itself. Thus, this issue should be addressed by the data perspective, in practical applications, when few-shot normal samples are insufficient to represent their class, we can increase the number of reference samples (or use the method we stated in the response to Reviewer 226V's Weakness 3). However, for method comparison, the few-shot setting is feasible (InCTRL also follows this setting). Because we ensure that all methods use the same reference samples, all methods can obtain the same representation information, the result comparison is reasonable and fair.
> > >
> > > Matching the nearest reference features is to generate residual features, InCTRL also employs this way to generate residuals (please see our response to Reviewer fL7E's Weakness 1), which also indicates that this way is reasonable and effective for residual generation. We think that besides residual features, the other parts of our method also play a crucial role, *e.g.* the Feature Constraintor and the Abnormal Invariant OCC loss (please see Table 2(a) in our paper). In addition, we note that in SPADE, each test feature will search for the nearest normal feature and directly calculate the distance between the two features as the anomaly score. By comparison, our method can significantly outperform SPADE, this also indicates that residual feature learning (v.s. only searching for the nearest normal features) is more effective for utilizing few-shot normal samples.

---

> > > ### Author Response · Authors · 2024-08-09
> > >
> > > **[To R2].** The results on the VisA dataset under the 4-shot setting and 2-shot setting are as follows (from Table 1 in the paper):
> > >
> > > | | RDAD | UniAD | SPADE | PaDiM | PatchCore | RegAD | ResAD |   &emsp; &emsp; | WinCLIP | InCTRL | ResAD$^{†}$ |
> > > | --- | --- | --- | --- | --- | --- | --- | --- | --- | --- | --- | --- |
> > > | 2-shot | 56.4/79.9 | 52.1/81.8 | 71.7/65.4 | 68.7/91.5 | 65.0/80.4 | 70.6/93.3 | 79.9/**96.4** | | 81.9/94.9 | **85.8**/- | 84.5/95.1|
> > > | 4-shot | 56.4/79.9 | 52.1/81.8 | 75.0/65.4 | 75.3/93.3 | 71.7/87.1 | 78.0/93.5 | 86.2/97.4 | | 84.1/95.2 | 87.7/- | **90.8**/**97.5**|
> > >
> > > We note that for a fair comparison, we should compare ResAD with its front methods (with the commonly used WideResNet50 as the feature extractor) and compare ResAD$^{†}$ with WinCLIP and InCTRL (all use ViT-B/16+ as the feature extractor). The results show that only the image-level AUROC under the 2-shot setting is lower than InCTRL, but all other results are better. Under the 4-shot setting, our ResAD$^{†}$ can significantly outperform InCTRL by 3.1\%. In addition, InCTRL only achieves image-level anomaly detection, while our method can achieve image-level anomaly detection and also pixel-level anomaly localization (please see our response to Reviewer fL7E's Weakness 1). Our ResAD$^{†}$ also significantly outperforms WinCLIP by 6.7\%/2.3\% under the 4-shot setting. The results of our response to Reviewer fL7E are as follows:
> > >
> > > | | FastRecon |ResAD | &emsp; &emsp; | AnomalyGPT | AnomalyGPT (ViT-B/16+) | InCTRL | ResAD$^{†}$ |
> > > | --- | --- | --- | --- | --- | --- | --- |--- |
> > > | VisA | 82.7/94.1 |86.2/97.4 | | **91.3**/88.8 | 85.4/86.9 | 88.7/- | 90.8/**97.5** |
> > >
> > > Although the image-level AUROC is slightly lower than AnomalyGPT by 0.5\% (please note that the original image encoder in AnomalyGPT is significantly larger than the ViT-B/16+ used in InCTRL and our ResAD$^{†}$), our method can significantly outperform AnomalyGPT by 5.4\%/10.6\% when using the same feature extractor (AnomalyGPT (VIT-B/16+) v.s. ResAD$^{†}$). Moreover, our method's pixel-level AUROC is significantly higher than AnomalyGPT.
> > >
> > > We think that evaluating the effectiveness of a method should not only focus on one dataset, but on multiple datasets. On the MVTec3D dataset, our method significantly outperforms other methods, the results are as follows.
> > >
> > > | | RDAD | UniAD | SPADE | PaDiM | PatchCore | RegAD | ResAD | &emsp; &emsp; | WinCLIP | InCTRL | ResAD$^{†}$ |
> > > | --- | --- | --- | --- | --- | --- | --- | --- | --- | --- | --- | --- |
> > > | 2-shot | 58.7/90.4 | 51.7/89.4 | 62.5/78.6 | 59.6/94.3 | 58.8/83.4 | 59.5/96.4 | 64.5/95.4 | | 74.1/96.8 | 68.9/- | **78.5**/**97.5**|
> > > | 4-shot | 58.7/90.4 | 51.7/89.4 | 62.3/78.6 | 62.8/94.5 | 61.5/87.1 | 62.3/96.7 | 70.9/97.3 | | 76.0/97.0 | 69.1/- | **82.4**/**97.9**|
> > >
> > > | | FastRecon | ResAD | &emsp; &emsp; | AnomalyGPT | AnomalyGPT (ViT-B/16+) | InCTRL | ResAD$^{†}$ (Ours) |
> > > | --- | ---| --- | --- | --- | --- | --- | --- |
> > > | MVTec3D | 66.5/95.2 | 70.9/97.3 | | 81.7/96.5 | 75.3/96.2 | 69.1/- | **82.4**/**97.9**|
> > >
> > > In our response to Weakness 1 of Reviewer CQaQ, we also further evaluate our method on a medical image dataset, BraTS and a video AD dataset, ShanghaiTech to validate the cross-domain generalization ability of our method. We list the results as follows (you can also see our response to Reviewer CQaQ's Weakness 1):
> > >
> > > | | RDAD | UniAD | SPADE | PaDiM | PatchCore | RegAD | ResAD | &emsp; &emsp; | WinCLIP | InCTRL | ResAD$^{†}$ |
> > > | --- | --- | --- | --- | --- | --- | --- | --- | --- | --- | --- | ---|
> > > | BraTS | 50.2/58.7 | 57.3/85.1 | 67.1/93.6 | 63.4/93.1 | 70.5/94.6 | 62.1/89.4 | 74.7/94.0 | | 68.9/93.5 | 76.9/- | **84.6**/**96.1**|
> > > | ShanghaiTech | 56.2/77.6 | 55.9/79.4 | 77.1/87.4 | 74.3/85.9 | 77.8/88.2 | 76.4/87.7 | 79.8/89.5 | | 79.6/88.6 | 69.2/- | **84.3**/**92.6**|
> > >
> > > The results show that when applied to medical images and video scenarios, the cross-domain generalization ability of our method is also more superior to other methods. Therefore, we think that based on the results of multiple datasets, our method is overall superior to other methods.
> > >
> > > Thank you again for your further response. We hope the above discussions can solve your concerns. If you still have any questions, we sincerely wish to further discuss with you.

---

> ### Comment · Reviewer_6bzj · 2024-08-11
>
> Thanks for your further response and provide more empirical results.  The response addresses most concerns.
> However, I noticed that the results were based on new medical datasets. How about the results on the headct and brainmri datasets demonstrated in InCTRL?
> I am curious whether the proposed ResAD is sensitive to the dataset.

---

> > ### Author Response · Authors · 2024-08-11
> >
> > We greatly appreciate your further response. We use the BraTS dataset because it provides ground-truth masks, while the BrainMRI and HeadCT datasets do not have pixel-level annotations, thus pixel-level AUROCs cannot be measured. Under the 2-shot and 4-shot settings, we further run our method and evaluate it on the BrainMRI and HeadCT datasets. The comparison (image-level AUROC) with InCTRL and also with the other results in its paper are as follows:
> >
> > | | SPADE | PaDiM | PatchCore | RegAD | ResAD | &emsp; | WinCLIP | InCTRL | ResAD$^{†}$ |
> > | ---| ---| ---| ---| ---| ---| ---| ---| ---| ---|
> > | BrainMRI(2-shot) | 75.4 | 65.7 | 70.6 | 44.9 | 92.3 | &emsp; | 93.4 | **97.3** | 97.0|
> > | BrainMRI(4-shot) | 75.9 | 79.2 | 79.4 | 57.1 | 93.8 | &emsp; | 94.1 | 97.5 | **97.9**|
> >
> > | | SPADE | PaDiM | PatchCore | RegAD | ResAD | &emsp; | WinCLIP | InCTRL | ResAD$^{†}$ |
> > | ---| ---| ---| ---| ---| ---| ---| ---| ---| ---|
> > | HeadCT(2-shot) | 64.5 | 59.5 | 73.6 | 60.2 | 90.4 | &emsp; | 91.5 | 92.9 | **93.5**|
> > | HeadCT(4-shot) | 62.4 | 62.2 | 80.5 | 52.2 | 91.7 | &emsp; | 91.2 | 93.3 | **94.6**|
> >
> > The results show that the generalization performance of our method on BrainMRI and HeadCT is also good. We hope the above response can solve your concerns. We sincerely wish to further discuss with you.

---

### Official Review · Reviewer_226V · 2024-07-12

**Soundness:** 4
**Presentation:** 4
**Contribution:** 3
**Rating:** 7
**Confidence:** 5

**Summary:**

This paper proposes a simple but effective framework that can be directly applied to detect anomalies in new classes. The main insight is learning the residual feature distribution rather than the initial one. In this way, we can significantly reduce feature variations. Even in new classes, the distribution of normal residual features would not remarkably shift from the learned distribution. Experiments were conducted on four datasets and achieved remarkable anomaly detection results.

**Strengths:**

The paper is original, high quality, clear, and easy to understand. The proposed method has a good heuristic effect on establishing a general anomaly detection model and will become a valuable baseline for the community after the release of the code.

**Weaknesses:**

1. Although unnecessary, I recommend punctuation at the end of a formula. This is one of the few formatting problems I can pick out. [Well written]
2. In Figure (b), it is suggested that abnormal should use a triangle icon. The difference between a hexagon and a circle is too small to see clearly.
3. The large difference between normal images should be considered, and image difference indicators such as FID and LPIPS can be used to calculate the difference inside the normal images in the data set you show. The difference should be relatively small, which is a potential false alarm hazard.
4. As stated in point 4 of the questions, the experimental setup of training on MVTecAD and then testing on the various classes of VisA is not reasonable.

**Questions:**

1. If the difference between normal images is relatively large, such as the breakfast_box class and screw_bag class in the MVTec Loco AD dataset, how can the reference feature pool be ensured? Intuitively, if the normal image difference is too large and the reference feature pool is not representative, the scheme has the hidden danger of a high false detection rate. If you're not using this dataset for an experiment, you should mention it in the text, which is good for the community.
2. Is the random selection of normal features in the reference feature pool a good strategy at fixed? Is it better to maximize the difference?
3. pixel-level AUROCs of InCTRL in Table 1 should be displayed. If not, it should be explained that it was not done by itself rather than that it could not be obtained.
4. Line 225: As far as I know there are 15 products and their corresponding exceptions in MVTecAD. Did you train the model using 15 product images and test it on various VisA classes? Although MVTecAD and VisA are two different data sets, they are just two sets containing multiple classes. So I think you should show the results of training with n classes from MVTecAD and testing on the remaining 15-n classes, with n as a hyperparameter looking for sensitivity, instead of testing across datasets. It would not be difficult for you to write your experiment in detail, and it would be more convincing if the results of the experiment appeared in the paper.

**Limitations:**

This paper objectively mentions the limitations of this article, and there is no potential negative impact.

---

> ### Author Rebuttal · Authors · 2024-08-05
>
> **[To W1].** Thanks for your suggestion. We then checked the formula writing in several other papers and found that there is punctuation at the end of a formula. This is a quite good detail suggestion. We will make modifications in the revised version.
>
> **[To W2].** Thanks for your suggestion. Using triangles to represent anomalies does look clearer. We will further improve Figure 1 in the revised version.
>
> **[To W3, Q1, and Q2].** We greatly appreciate your suggestion. Yes, when the difference between normal images is too large, it may cause the reference feature pool is not representative. For practical applications, this issue should be particularly focused and reasonably addressed. Of course, the simplest resolution is to increase the number of reference samples. This is feasible, as in practical applications, the number of reference samples is usually not as strict as the 2-shot and 4-shot (following previous papers) in our paper. From the perspective of method comparison, we think that random selection is ok, as long as we ensure that all methods use the same reference samples, the result comparison is reasonable. **[To Q2]** However, in practical applications, we expect that the reference samples can fully represent their class, so it's best to have sufficient differences between the reference samples. Thus, the sample selection strategy cannot be random. **[To Q1]** A feasible method is to first cluster all available normal samples into different clusters based on a clustering algorithm (*e.g.*, KMeans). Then, based on the number of reference samples, we evenly distribute it to each cluster. When selecting from a cluster, we can prioritize selecting samples closer to the center. During clustering, we think that the FID and LPIPS you mentioned are good ways to calculate the difference between two samples. In addition, when there are a large number of reference samples, we can also use the method in PatchCore to select coreset features as reference features, which will be more efficient and also representative. In the revision, we will add the above discussion on sample selection strategy to the paper.
>
> **[To W4 and Q4].** Yes, we train the model using 15 classes in MVTecAD and test it on 12 classes in VisA. For MVTecAD, we train on 12 classes in VisA and test on 15 classes in MVTecAD. Adopting the cross-dataset experimental setup is because we think that it's more challenging than cross-classes within a single dataset, thus can better verify the model's class generalization ability. A dataset may be collected under the same photography condition, variations in other factors besides the object itself may be minimal. For example, all images in MVTecAD only contain a single object, while the images of some classes in VisA have multiple objects. The backgrounds in MVTec3D are all black, which is not the case in other datasets. In addition, we also follow InCTRL to use the cross-dataset experimental setup.
>
> We think that your mentioned experimental setting is also very reasonable, as by varying $n$, we can demonstrate the sensitivity of the model to different numbers of training classes. Under the 4-shot setting, we further run our model on MVTecAD (with $n=5, 10$). Note that different $n$ means the number of test classes $15-n$ is different (this will cause the test results of different n cannot be compared with each other). Thus, we use fixed 5 classes as the test classes, including hazelnut, pill, tile, carpet, and zipper. For $n=5$, the training classes include bottle, cable, capsule, grid, and leather. For $n=10$, the training classes include bottle, cable, capsule, grid, leather, metal nut, screw, toothbrush, transistor, and wood. The results are as follows:
>
> |n=5 | n=10 | VisA to MVTecAD |
> | --- | --- | --- |
> | 96.4/97.6 | 96.8/97.9 | 95.1/97.2 |
>
> The results demonstrate that cross-dataset is more challenging than cross-class in a single dataset. With more training classes, the results will be better, but the model is not very sensitive. Due to too many experiments, we currently do not have enough time to provide the results of other methods under this setting. In the revision, we will also add this setting to our experimental setup and complete other methods' results.
>
> **[To Q3].** For each input image, InCTRL finally only outputs an image-level anomaly score. Thus, InCTRL cannot calculate pixel-level AUROCs. In the revision, we will add a relevant explanation.
>
> If you still have any questions, we are very glad to further discuss with you.

---

> > ### Comment · Reviewer_226V · 2024-08-11
> >
> > 1. The paper’s logic is clear and the discussion is sufficient. After the code is released, the method can become a highly referenced article in class generalizable anomaly detection.
> > 2. the author's reply on the reference feature pool is satisfactory. It is difficult to fully answer this question experimentally during the rebuttal period, still, I hope the author will add the above discussion on sample selection strategy to the paper, as you said, to promote the development of community research.
> > 3. The author's improvement on the evaluation protocol makes me believe in the code basis of his work, and his grasp of the whole experimental design is reasonable and confident.

---

> > > ### Author Response · Authors · 2024-08-11
> > >
> > > We greatly appreciate your further response. We will add the discussions from the rebuttal response to the revised paper and also release the code.

---

### Official Review · Reviewer_CQaQ · 2024-07-12

**Soundness:** 3
**Presentation:** 3
**Contribution:** 4
**Rating:** 6
**Confidence:** 4

**Summary:**

This paper proposed a simple yet effective framework ResAD for class-generalizable anomaly detection by leveraging residual feature learning and a hypersphere constraint. The framework's ability to generalize to new classes without retraining or fine-tuning makes it valuable for real-world applications, providing significant improvements over existing methods. Comprehensive experiments on four real-world industrial AD datasets (MVTecAD, VisA, BTAD, and MVTec3D) demonstrate ResAD's superior performance.

**Strengths:**

(1)ResAD effectively addresses the challenge of class-generalizable anomaly detection, the generalization ability using only a few normal samples as references makes it highly practical for real-world applications.

(2)The use of residual feature learning to reduce feature variations and improve generalizability is novel and effective

(3)The approach is shown to be robust across different datasets and settings.

**Weaknesses:**

(1)The experiments are primarily conducted on industrial anomaly detection datasets. While these are relevant, the method's generalizability to other domains, such as medical images or video data, is not fully explored.

(2)The selection of few-shot reference samples may impact performance. Previous methods typically run multiple independent runs using different random seeds to ensure robustness. However, this work only provides results from a single group of samples, which may not fully represent the model's performance variability.

**Questions:**

(1)If the few-shot reference samples contain anomalies,  will it impact the overall performance a lot?

(2)The way to combine residual feature learning with existing models is not explicitly defined.

**Limitations:**

Limitations are discussed in Appendix Sec. B.

---

> ### Author Rebuttal · Authors · 2024-08-05
>
> **[To W1].** We greatly appreciate your suggestion. Under the 4-shot setting, we further evaluate our method on a medical image dataset, BraTS (for brain tumor segmentation) and a video AD dataset, ShanghaiTech (as our method is image-based, we extract video frames as images for use). The comparison results are as follows:
>
> | | RDAD | UniAD | SPADE | PaDiM | PatchCore | RegAD | ResAD | WinCLIP | InCTRL | ResAD$^{†}$|
> | --- | --- | --- | --- | --- | --- | --- | --- | --- | --- | --- |
> |BraTS |50.2/58.7 | 57.3/85.1 | 67.1/93.6 | 63.4/93.1 | 70.5/94.6 | 62.1/89.4 | 74.7/94.0 | 68.9/93.5 | 76.9/- | 84.6/96.1|
> |ShanghaiTech|56.2/77.6 | 55.9/79.4 | 77.1/87.4 | 74.3/85.9 | 77.8/88.2 | 76.4/87.7 | 79.8/89.5 | 79.6/88.6 | 69.2/- | 84.3/92.6|
>
> The results show that when applied to medical images and video scenarios, the cross-domain generalization ability of our method is also good. In the revision, we will include the above results and also results under the 2-shot setting in our paper. We will also attempt more datasets and further discuss our method's generalizability to other domains.
>
> **[To W2].** We greatly appreciate your suggestion. Although we only used a single group of few-shot samples, we strictly ensured that all methods used the same few-shot samples. So, when writing the paper, we thought that the result comparison was reasonable and didn't run on other groups. After reading your suggestion, we think that results on multiple groups are necessary. We then randomly select two groups of few-shot samples and obtain the following results:
>
> | | Group1 | Group2 | Results in paper | Mean$\pm$Std|
> | --- | --- | --- | --- | --- |
> |MVTecAD | 91.0/96.0 | 90.7/95.9 | 90.5/95.7 | 90.7$\pm$0.21/95.9$\pm$0.12|
> |VisA | 86.3/97.5 | 86.9/97.6 | 86.2/97.4 | 86.5$\pm$0.31/97.5$\pm$0.08|
> |BTAD | 95.3/97.5 | 95.4/97.6 | 95.6/97.6 | 95.4$\pm$0.12/97.6$\pm$0.05|
> |MVTec3D | 70.2/97.1 | 70.5/97.3 | 70.9/97.3 | 70.5$\pm$0.29/97.2$\pm$0.09|
>
> In the revision, we will include the above results and also results under the 2-shot setting in our paper. Due to too many experiments, we currently do not have enough time to provide the results of other methods on these two groups. We will further supply these results in the revised version.
>
>
> **[To Q1].** We think that it will have an impact on the performance. Because it may cause some abnormal features to match similar abnormal features from the reference samples, making these abnormal residual features hard to distinguish from normal residual features. Since the few-shot samples are used as reference, it should be natural not to contain anomalies. For real-world applications, it's also not very hard for us to ensure that the few-shot reference samples are all normal.
>
> **[To Q2].** As UniAD and RDAD are both feature-based AD methods, combining our residual feature learning with them is straightforward. For UniAD, we convert the initial features into residual features and then perform subsequent feature reconstruction. For RDAD, the initial features extracted by the teacher network are converted into residual features as the learning target of the student network. Then, the student network is trained to predict residual representations of the teacher network. In the revision, we will add relevant details to describe clearly how to combine residual feature learning with existing models.
>
> If you still have any questions, we are very glad to further discuss with you.

---

> > ### Comment · Reviewer_CQaQ · 2024-08-13
> >
> > Thanks for the authors' response, and it addressed my concerns. I will maintain my score.

---

### Official Review · Reviewer_fL7E · 2024-07-24

**Soundness:** 2
**Presentation:** 2
**Contribution:** 2
**Rating:** 5
**Confidence:** 5

**Summary:**

This paper proposes to address cross-class anomaly detection problem. To this end, this study introduce a residual learning framework ResAD. The ResAD framework aims to learning residual feature distribution between target image and reference image. Experiments are conducted to valid the effectiveness of the proposed method.

**Strengths:**

1. The cross-class/class-generalize anomaly detection is a crutial task in the realm of anomaly detection.
2. The structure of ResAD is simple and effective.

**Weaknesses:**

1. The idea of residual estimation is highly similar to InCTRL [1].
2. Lack of comparision with FastRecon[2] and AnomalyGPT[3].
3. The writing should be improved. The optimization terms are unclear and hard to follow.
4. In Table.5, there is a reproduced result of WinCLIP on WideResNet50, however the windows in WinCLIP is designed for VIT, how can the authors report the result?


[1] Jiawen Zhu and Guansong Pang. Toward generalist anomaly detection via in-context residual learning with few-shot sample prompts. In CVPR, 2024.
[2] Fang Z, Wang X, Li H, et al. Fastrecon: Few-shot industrial anomaly detection via fast feature reconstruction[C]//Proceedings of the IEEE/CVF International Conference on Computer Vision. 2023: 17481-17490.
[3] Gu Z, Zhu B, Zhu G, et al. Anomalygpt: Detecting industrial anomalies using large vision-language models[C]//Proceedings of the AAAI Conference on Artificial Intelligence. 2024, 38(3): 1932-1940.

**Questions:**

1. The OCC loss is used for constrain the distribution to a fixed region and the NF model is used for estimate the distribution. If the distribution is fixed, what is the meaning of NF model?
2. There are three loss terms, how about the sensitivities of the balance between the three loss terms.
3. There is no ablation study to valid the effectiveness of the proposed loss terms which makes the method lack credibility.

---

> ### Author Rebuttal · Authors · 2024-08-05
>
> **[To W1].** Thanks for your professional review. We think that our work and InCTRL should be concurrent work. We also initially submitted our work to CVPR2024, and received two weak accepts and one reject (you can see the relevant materials in the rebuttal pdf file). Regretfully, our work was rejected at that time. So, we should have proposed the idea of residual learning independently of InCTRL and almost at the same time. But our method has obvious differences with InCTRL in the definition and utilization of residuals. This (*i.e.*, two independent works almost simultaneously proposed the residual learning idea) also demonstrates residual learning is an effective way to achieve class-generalizable anomaly detection. Then, we made comprehensive revisions to our paper based on the reviewers' suggestions. Subsequently, we also noticed the InCTRL paper and felt that our method has advantages compared to it. Thus, we submitted our revised paper to NeurIPS2024. In CVPR2024, our paper was rejected mainly because a reviewer thought our result comparison with unsupervised AD methods was unreasonable and unfair. In this submission, we mainly compare with few-shot AD methods and InCTRL. The main differences between our method and InCTRL are as follows:
>
> (1) The definition of residuals in InCTRL is based on feature distances. The residual map is defined by ((E.q.(1) in the InCTRL paper): $M_x^l(i,j) = 1 - \left<T_x^l(i,j),h(T_x^l(i,j)|\mathcal{P}^\prime)\right>$, where $h(T_x^l(i,j)|\mathcal{P}^\prime)$ returns the embedding of the patch token that is most similar to $T_x^l(i,j)$ among all image patches in $\mathcal{P}^\prime$, and $\left<\cdot\right>$ is the cosine similarity function. Thus, InCTRL is based on residual distance maps, while our method is based on residual features.
>
> By comparison, we think that residual distances in InCTRL can limit the range of residual representation (as the cosine similarity is in [-1,1]). This is not beneficial for distinguishing between normal and abnormal regions, as a position on the residual map is only represented by a residual distance value. Within a limited representation range (1-[-1,1] $\rightarrow$ [0,2]), normal and abnormal residual distance values are more likely to be not strictly separable. Thus, for a position on the residual map, it's hard for us to make decision based on a scalar value. So, InCTRL makes image-level classification based on a whole residual map (see the following (2)). In contrast, our residual features don't limit the range of residual representation and can retain the feature properties. In high-dimensional feature space, we can also establish better decision boundaries between normal and abnormal (a basic idea in machine learning: solving low-dimensional inseparability by converting to high-dimension).
>
> (2) InCTRL devises a holistic anomaly scoring function $\phi$ to learn the residual distance map $M_x = \frac{1}{n}\sum_{l=1}^{n}M_x^l$ and convert it to an anomaly score: $s(x) = \phi(M_x^{+};\Theta_\phi) + \alpha s_p(x)$ (E.q.(8) in the InCTRL paper), where $M_x^{+} = M_x \oplus s_i(x) \oplus s_a(x)$ (E.q.(7)). $s_i(x)$ is an anomaly score based on an image-level residual map $F_x$ (see E.q.(4) in the InCTRL paper) and $s_a(x)$ is a text prompt-based anomaly score. Thus, InCTRL is to train a binary classification network based on residual distance maps. For each input image, InCTRL finally only outputs an image-level anomaly score. Our method is to learn the distribution of residual features, an anomaly score can be estimated for each feature, thus can be used to locate anomalies.
>
> (3) Due to the designs in InCTRL that we mentioned above, one main advantage of our method is that it can achieve image-level anomaly detection and also pixel-level anomaly localization, while InCTRL only achieves image-level anomaly detection functionality.
>
> As for performance, the average results on four AD datasets of our method are better than InCTRL's (please see Table 1 in our paper). In our response to Weakness 1 of Reviewer CQaQ, the cross-domain generalization ability of our method is also better.
>
> **[To W2].** We greatly appreciate your suggestion. Under the 4-shot setting, we further reproduce FastRecon and AnomalyGPT by using the same few-shot samples as ours. Based on the official open-source code, we obtain the following results:
>
>
> |    | FastRecon | AnomalyGPT | AnomalyGPT (ViT-B/16+) | InCTRL | ResAD$^{†}$ (Ours) |
> | ----------- | ----------- | ----------- | ----------- |----------- | ----------- |
> | MVTecAD | 92.3/96.2 | 95.0/96.0 | 92.1/95.3 | 94.5/- | 94.2/96.9|
> |VisA | 82.7/94.1 | 91.3/88.8 | 85.4/86.9 | 88.7/- | 90.8/97.5 |
> |BTAD | 90.1/93.4 | 92.0/96.0 | 90.2/94.9 | 91.7/- | 91.5/96.8 |
> |MVTec3D | 66.5/95.2 | 81.7/96.5 | 75.3/96.2 | 69.1/- | 82.4/97.9|
> |**Average**| 82.9/94.7 | 90.0/94.3 | 85.8/93.3 | 85.8/- | 89.7/97.3|
>
> Note that the original image encoder in AnomalyGPT is significantly larger than the ViT-B/16+ used in InCTRL and our ResAD$^{†}$. When AnomalyGPT also uses ViT-B/16+ as the image encoder, our method is more superior to it.
> In the revision, we will include the above results and also results under the 2-shot setting in our paper. We will also cite FastRecon and AnomalyGPT and add discussions with them.
>
> **[To W3].** Thanks for your suggestion. We will carefully check and modify obscure expressions and unclear concepts to make our paper easier to follow. We will also seriously consider other reviewers' suggestions to further improve the quality of our paper.
>
> **[Others].** For Weakness 4 and questions, please see the rebuttal pdf file or the following Comment.
>
> If you still have any questions, we are very glad to further discuss with you.

---

> ### Author Response · Authors · 2024-08-05
>
> **[To W4].** The window mechanism of WinCLIP is not limited to ViT. In WinCLIP, the window mechanism is mainly used to address the issue that the local patch features extracted by the CLIP image encoder are not aligned with the text features. The window mechanism provides image patches of different scales, which are then sent into the CLIP image encoder to obtain global features that can align with the text features. The window mechanism is intrinsically similar to the classic sliding window in object detection, thus can also be used in CNN networks. We can send image patches provided by the window mechanism into WideResNet50, and also obtain the window embedding maps of different scales as shown in Figure 4 of the WinCLIP paper. However, because the features of WideResNet50 are not aligned with the text features, we remove the language-guided anomaly score map and only generate the vision-based anomaly score map based on the few-shot normal samples (the WinCLIP+ in the WinCLIP paper). In lines 516-518, we have briefly mentioned our adaptation way. In the revision, we will add more details about the experiments in Table 5.
>
> **[To Q1].** The goal of our Feature Constraintor (optimized by the OCC loss) is to constrain initial residual features to a spatial hypersphere for further reducing feature variations. After the Feature Constraintor, feature variations can effectively be reduced, but this does not mean that the feature distribution is fixed within the hypersphere. Moreover, even if the feature distribution is fixed, we still need to learn the distribution and then can perform anomaly detection based on the learned distribution. The NF model is used to learn the feature distribution.
>
> **[To Q2].** During training, we found that summing up the three loss terms and then backpropagating gradients to optimize the whole model would lead to unstable training. Then, we used the "torch.detach()'' method in the Pytorch library to detach the features after the Feature Constraintor and then sent the detached features into the NF model. This simple way can make the model training more stable. In the revision, we will add this code detail to the paper. Thus, the weight of $L_{occ}$ can be set as 1 (*i.e.*, we can not need to balance $L_{occ}$ with $L_{ml}$ and $L_{bg-spp}$, as the Feature Constraintor and the NF model parts are separated in the gradient graph). When training the NF model, the $L_{ml}$ is the basic loss. So, we keep the weight of $L_{ml}$ as 1 and set a variable $\lambda$ as the weight of $L_{bg-spp}$. By varying different $\lambda$ values, the results (under the 4-shot setting) about the sensitivity are as follows:
>
> | $\lambda$ | 0.1 | 0.5 | 1 | 2 | 3 | 5 | 10 |
> | --- | --- |--- |--- |--- |--- |--- |--- |
> |  |89.1/94.9 | 90.0/95.6 | 90.5/95.7 | 90.6/96.0 | 89.8/95.3 | 89.3/95.2 | 88.7/94.9|
>
> Both small and large $\lambda$ values can lead to performance degradation. $L_{bp-spp}$ is to assist model in learning abnormal residual features. Small $\lambda$ may cause the impact of abnormal features on final loss to be relatively small. Large $\lambda$ may lead to overfitting to known anomalies, which is not conducive to generalization.
>
> **[To Q3].** In our paper, we mainly propose the Abnormal Invariant OCC (AI-OCC) loss $L_{occ}$. The maximum likelihood loss $L_{ml}$ is the basic loss for training the NF model. The $L_{bg-spp}$ loss is following BGAD to effectively utilize abnormal features. In fact, the framework ablation studies in Table 2(a) also include ablation studies regarding our proposed AI-OCC loss. In Table 2(a), “w/o Feature Constraintor'' means the $L_{occ}$ is not used (as the Feature Constraintor is removed), “w/o Abnormal Invariant OCC loss'' means the $L_{occ}$ only has the first part of E.q.(2), while is not our proposed AI-OCC loss. In lines 293-298, we also provide discussions about the ablation studies of our AI-OCC loss.
>
> **[Limitations].** In Appendix Sec.B, we provide discussions about our method's limitations. In the revision, we will further discuss the limitations of our method more comprehensively and clearly based on the comments of all reviewers.

---

> ### Comment · Reviewer_fL7E · 2024-08-12
> **Response**
>
> The authors' rebuttal has addressed most of my concerns and I believe that the generalization ability of the proposed method is practical and significant, so I decide to raise my score.  However, I still have some concerns about the conflicts of OCC and NF. Also, I observe the rebuttal file in the supplementary (Line 42-51) that same problem have mentioned by other comments, so I think it would be a major concern about the method. It would be better If there are some more convicing validation.

---

> > ### Author Response · Authors · 2024-08-12
> >
> > We greatly appreciate your further response. We think that your concern should be whether both the Feature Constraintor (namely the OCC part) and the NF model are necessary, namely, whether we can achieve anomaly detection only based on the OCC part, or whether we only need the NF model without the OCC part. We think the ablation study (``w/o Feature Constraintor'', namely only using the NF model) in Table 2(a) can answer the latter question. When we design the Feature Constraintor, our motivation is to employ it to constrain initial residual features to a spatial hypersphere for further reducing feature variations. In this way, for the features sent into the NF model, the distribution of new classes could be more consistent with the learned distribution. Thus, it's beneficial to achieve better cross-class AD results. After reading your response, we realize that we need to design an ablation study in which we directly perform anomaly detection based on the output of the Feature Constraintor without the NF model. Specifically, we regard the OCC part as a conventional OCC-based AD model and utilize the distances from the features to the OCC center as the anomaly scores (this is also a commonly used way for measuring anomaly scores in OCC-based AD methods). Under the 4-shot setting, we obtain the following results (also the results from the paper):
> >
> > | | w/o Feature Constraintor | Only Feature Constraintor | ResAD |
> >  | --- | --- | --- | --- |
> > | From VisA to MVTecAD | 82.3/93.5 | 82.9/89.3 | 90.5/95.7 |
> >
> > The results show that the OCC part and the NF model don't have conflicts. As you mentioned in Question 1, the ideal situation is that even in new classes, normal feature distribution is fixed within a hypersphere, while all anomalous features are outside the hypersphere. Then, only the OCC part is enough to achieve good AD results. However, in practical optimization, it's hard to achieve the ideal situation. After the Feature Constraintor, normal and abnormal features may still not be fully separable based on the distances. Then, the NF model is used to learn the feature distribution, which can assist us in better distinguishing normal and abnormal features. Therefore, we think that further learning the feature distribution after the OCC part is beneficial. In the revision, we will add the new ablation study and the relevant discussions to our paper.
> >
> > We hope the above discussions can solve your concerns. If not, we are very glad to further discuss with you.

---

### Author Rebuttal · Authors · 2024-08-05

We are very grateful for all your constructive suggestions. Please see our specific responses to each reviewer.

In the Author Rebuttal pdf file, we provide some relevant materials. We recommend that Reviewer fL7E and 6bzj can download the pdf file and see the contents in it.

---

### Decision · Program_Chairs · 2024-09-25

**Decision:**

Accept (spotlight)

**Comment:**

The work tackles a problem called class-generalizable anomaly detection on image data and introduces a new approach called ResAD for the problem. The key idea in the approach is residual feature learning based on a popular VLM backbone, CLIP. Th approach is evaluated on four industrial defect detection datasets and compared with several relevant methods, including two recent CLIP-based methods, WinCLIP and InCTRL.

After the author-reviewer discussion period, all reviews are positive toward this work. Reviewers CQaQ and 226V are inclined to acceptance throughout the review process. Reviewer 6bz had some major doubts on this work in the model design and the experimental comparison. The authors managed to address these concerns properly, resulting in an increase of rating from borderline accept to weak accept. Reviewer fL7E had major concerns over the difference of the work to a recent method InCTRL and comparison to related methods FastRecon and AnomalyGPT. The authors have also adequately addressed these concerns during the author-reviewer discussion period, leading to a rating increase from borderline reject to borderline accept.

Overall, all reviewers generally appreciate this work in terms of problem setting, technical design, and empirical justification. I therefore recommend accepting the paper to NeurIPS.

I'd like to further recommend the paper as a spotlight paper because i) the studied problem is an emerging important AD setting that uses large pre-trained vision-language models to support class-generalizable AD (or generalist AD), ii) the proposed residual feature learning idea is interesting (despite a similar insight was proposed in InCTRL), and iii) the detection performance is promising.